# `floq`: Training Critics via Flow-Matching for Scaling Compute in Value-Based RL

**Bhavya Agrawalla**
bbagrawa@andrew.cmu.edu

**Khush Agrawal**
khusha@andrew.cmu.edu

**Michal Nauman**
nauman.mic@gmail.com

**Aviral Kumar**
aviralku@andrew.cmu.edu

## Abstract

A hallmark of modern generative models is their reliance on training objectives that construct the target output iteratively, with dense supervision provided at intermediate steps, e.g., teacher forcing the next token in language models or step-by-step denoising in diffusion models. Such objectives allow models to capture complex functions in a broadly generalizable way. Motivated by this observation, we study the benefits of iterative computation for temporal difference (TD) methods in reinforcement learning (RL). Typically they represent value functions in a monolithic fashion, without iterative compute. We introduce `floq` (*flow-matching Q-functions*), an approach that parameterizes the Q-function using a velocity field and trains it using techniques from flow-matching. This velocity field underneath the flow is trained using a TD-learning objective, which bootstraps from Q-values produced by a target velocity field, computed by running multiple steps of numerical integration. Crucially, `floq` allows for more fine-grained control and scaling of the Q-function capacity than monolithic architectures, by appropriately setting the number of integration steps. Across a suite of challenging offline RL benchmarks and online fine-tuning tasks, `floq` improves performance by nearly $1.8\times$. `floq` scales capacity far better than standard TD-learning architectures, highlighting the potential of iterative computation for value learning.

**Code and runs for `floq` can be found at**: https://github.com/CMU-AIRe/floq

## 1 Introduction

A key principle in building effective models in various areas of machine learning is the use of ***iterative computation***: producing complex output functions by composing a sequence of simpler operations. E.g., language models based on transformers (Vaswani et al., 2017) can generate coherent text by predicting the next token or by composing atomic reasoning strategies (Gandhi et al., 2025). Similarly, diffusion and flow-matching models (Ho et al., 2020; Sohl-Dickstein et al., 2015; Lipman et al., 2023; Albergo & Vanden-Eijnden, 2023) synthesize images by progressively denoising small perturbations. Effective results from these models suggests that iterative computation is a powerful tool for modeling complex functions with deep networks, by scaling compute appropriately.

Motivated by these results, in this paper, we ask: ***can iterative computation also improve value estimation in reinforcement learning (RL)?*** Specifically, we are interested in improving the estimation of the Q-value function. While Q-functions map state-action inputs to a scalar value, they are known to be highly complex and difficult to fit accurately (e.g., (Dong et al., 2020)). Standard temporal-difference (TD) learning used to train Q-functions struggles to leverage capacity of deep networks (Kumar et al., 2021; 2022; Bjorck et al., 2021; Lyle et al., 2022; Gulcehre et al., 2022), often resulting in poor generalization. These problems are further exacerbated in the offline RL problem setting (Levine et al., 2020; Kumar et al., 2019), where we must learn entirely from static datasets. This motivates exploring architectures that spend compute iteratively to estimate value functions, potentially yielding better Q-values and policies.

A natural starting point for using iterative compute in value-based RL is to utilize a ResNet (He et al., 2016) Q-function, where stacking more residual blocks provides a way to run iterative computation. Recent work has obtained modest gains with ResNets (Kumar et al., 2023a;b; Farebrother et al., 2024; Nauman et al., 2024), but these methods need normalization and regularizers to enable stable training (Bjorck et al., 2021; Nauman et al., 2024; Lee et al., 2024; Kumar et al., 2023a). Despite improvements, these approaches lack one ingredient that makes iterative computation effective in transformers or diffusion models: *supervision at every step* of the iterative process. Just as next-token prediction supervises each generated token and diffusion supervises each denoising step, we hypothesize that stepwise loss supervision applied to TD learning might lead to improvements.

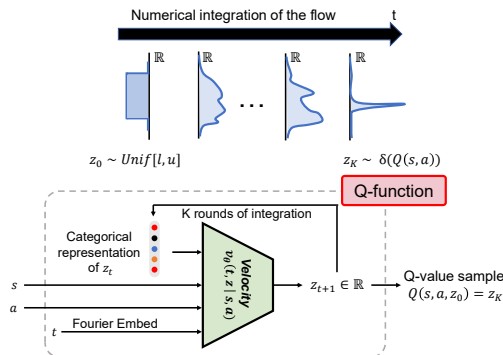

Figure 1: `floq` **architecture.** We model the Q-function via a velocity field of a flow-matching model. Over multiple calls, this velocity field converts a randomly sampled input $\boldsymbol{z}(0)$ into a sample from the Dirac-delta distribution centered at the mean Q-value. We build a flow-matching loss for training. Doing this enables us to scale computation by running numerical integration, with multiple calls to the velocity field. To train `floq`, we utilize a categorical representation of input $\boldsymbol{z}_t$ (Farebrother et al., 2024) and a Fourier representation of $t$.

With this observation, to effectively leverage iterative computation with dense supervision, we design a novel architecture for parameterizing Q-functions. Instead of using a single monolithic network, we represent the Q-function as a *velocity field* over a scalar value (Figure 1). Our approach, `floq` (flow-matching Q-functions) samples a scalar uniformly distributed noise and maps it to the Q-value by numerically integrating the predictions of the velocity field. We train the velocity with a linear flow-matching objective (Albergo & Vanden-Eijnden, 2023; Lipman et al., 2023), supervised to match the evolving TD-targets. At each step, we minimize the deviation between the current Q-value estimate and the corresponding TD-target. We introduce several design choices that stabilize training and help the architecture scale capacity effectively. These include appropriately setting the support of initial noise, using a categorical representation to handle non-stationary inputs and a Fourier time embedding to allow the velocity predictions to vary meaningfully across integration steps (Figure 2).

We use `floq` to represent the Q-function for a number of complex RL (Levine et al., 2020; Kumar et al., 2019) tasks from the OGBench (Park et al., 2025a) benchmark, previously studied by Park et al. (2025d). In aggregate, we find that `floq` outperforms offline RL algorithms that represent Q-functions using a monolithic network by nearly $1.8\times$. `floq` is superior even when these approaches are provided with more parameters, and more complex and higher capacity architectures. `floq` also outperforms existing methods when running online fine-tuning after offline RL pre-training. We also show that increasing the number of flow-matching steps results in better downstream policy performance. Allocating the same capacity via Q-network ensembles or ResNets performs worse.

## 2 RELATED WORK

**Expressive generative models in RL.** The most typical use of conventional generative models in RL has been to represent the policy, with several adoptions of diffusion policies (Wang et al.; Hansen-Estruch et al., 2023; Yang et al., 2023; Bansal et al., 2023; Li et al., 2024; Ren et al., 2024), flow-based policies (Lipman et al., 2023; Albergo & Vanden-Eijnden, 2023; Park et al., 2025d), and sequence policies (Janner et al., 2021; Lee et al., 2022; Yamagata et al., 2023). This shift is motivated by evidence that policy learning is often a significant bottleneck in offline RL (Kostrikov et al.; Park et al., 2024). In parallel, policy-agnostic frameworks such as PA-RL (Mark et al., 2024) decouple algorithmic progress from specific architectural choices, enabling the use of diffusion, flows, or transformers interchangeably. Complementarily, we do not focus on policy expressivity and instead aim to utilize more expressive Q-functions, and opt to study `floq` on top of FQL for simplicity.

**Scaling Q-functions.** Efforts to scale Q-functions in RL have taken multiple directions with new training objectives such as classification losses (Kumar et al., 2023a; Farebrother et al., 2024; Nauman et al., 2025; Seo et al., 2025), architectures (He et al., 2016; Kumar et al., 2023b; Chebotar et al., 2023; Obando-Ceron et al., 2024), and regularization strategies (Kumar et al., 2021; Lyle et al., 2021; Kumar et al., 2022; Nauman et al., 2024; Bhatt et al., 2024),. Previous work has also attempted to develop scaling laws for TD learning (Rybkin et al., 2025; Fu et al., 2025) and showing that alternatives to TD

can scale to deeper architectures (Wang et al., 2025). Despite these advances, a clear recipe for scaling value-based RL with TD-learning has yet to emerge. Our work demonstrates that compute-efficient scaling can be realized not simply by increasing depth or width, but by introducing dense intermediate supervision through multiple integration steps of the Q-function. `floq` introduces a novel axis of scaling, allowing for compute scaling through additional integration steps rather than depth or width.

**Scaling inference compute.** A complementary line of work studies how more inference-time compute can be traded for performance. Classical MPC-style planners coupled with learned dynamics models such as PETS (Chua et al., 2018), MPPI (Williams et al., 2017), and PDDM (Nagabandi et al., 2018) naturally allow scaling. In offline RL, MBOP (Argenson & Dulac-Arnold, 2020) explicitly adopts planning with a learned model, a behavior prior, and a terminal value to extend the effective horizon. Generative world models enable similar test-time scaling by planning inside the learned model (Janner et al., 2021; 2022). Similarly, performance of MCTS-style methods improves with more simulation (Schrittwieser et al., 2020; Hubert et al., 2021; Danihelka et al., 2022; Ye et al., 2021). Across all methods listed in this paragraph, the general pattern is that increasing test-time budget (i.e. simulations, horizon, candidate trajectories) improves returns up to the limit set by model bias and value estimation error. However, none of these works use more test-time compute to better estimate a value function. Our results show that `floq` can not only use more integration steps at inference time to amplify the "capacity" of the Q-function, but also that doing so during training helps us learn better Q-functions in the first place. We are the first to show that using more integration steps is a viable and effective path to scaling compute for critic networks.

## 3 PRELIMINARIES AND NOTATION

The goal in RL is to learn the optimal policy $\pi : \mathcal{S} \mapsto \mathcal{A}$ for an MDP $\mathcal{M} = (\mathcal{S}, \mathcal{A}, P, r, \rho, \gamma)$ that maximizes cumulative discounted value function, denoted by $V^\pi(s) = \sum_t \mathbb{E}_{a_t \sim \pi(s_t)} [\gamma^t r(s_t, a_t)|s_0 = s]$. $\mathcal{S}, \mathcal{A}$ denote the state and action spaces. $P(s'|s, a)$ and $r(s, a)$ are the dynamics and reward functions. $\rho(s)$ denotes the initial state distribution and $\gamma \in (0, 1)$ denotes the discount factor. The Q-function of a policy $\pi$ is defined as $Q^\pi(s, a) = \sum_t \mathbb{E}_{a_t \sim \pi(s_t)} [\gamma^t r(s_t, a_t)|s_0 = s, a_0 = a]$, and we use $Q_\theta^\pi$ to denote the estimate of the Q-function of a policy $\pi$ as obtained via a neural net with parameters $\theta$. Value-based RL methods train a Q-network by minimizing the temporal difference (TD) error:

$$L(\theta) = \mathbb{E}_{(s,a,s')\sim\mathcal{D},a'\sim\pi(\cdot|s')} \left[ \left( r(s, a) + \gamma \bar{Q}(s', a') - Q_\theta(s, a) \right)^2 \right], \quad (3.1)$$

where $\mathcal{D}$ is the offline dataset, $\bar{Q}$ is the target Q-network, $s$ denotes a state, and $a'$ is an action from policy $\pi(\cdot|s)$ that aims to maximize $Q_\theta(s, a)$. Offline RL methods are discussed in Appendix A.1.

**Flow-matching.** Given a target data distribution $p(\boldsymbol{x})$ over $\boldsymbol{x} \in \mathbb{R}^d$, flow-matching (Lipman et al., 2023; Liu et al., 2023; Albergo & Vanden-Eijnden, 2023) attempts to fit a time-dependent **velocity field**, $v_\theta(t, \boldsymbol{x}) : [0, 1] \times \mathbb{R}^d \to \mathbb{R}^d$ such that the solution $\psi_\theta(t, \boldsymbol{x})$ to the ODE: $\frac{d}{dt}\psi_\theta(t, \boldsymbol{x}) = v_\theta(t, \psi_\theta(t, \boldsymbol{x})), \psi_\theta(0, \boldsymbol{x}(0)) = \boldsymbol{x}(0)$ transforms samples $\boldsymbol{x}(0)$ from a simple base distribution (e.g., standard Gaussian or uniform, as we consider in this work) into samples from $p(\boldsymbol{x})$ at time $t = 1$. The most widely used approach is linear flow matching (Lipman et al., 2023), which trains the velocity flow to predict the gradient obtained along the linear interpolating path between $\boldsymbol{x}(0)$ and $\boldsymbol{x}(1)$ at all intermediate points. Concretely, define $\boldsymbol{x}(0) \sim p_0(\boldsymbol{x})$ be a sample from a simple initial distribution, $\boldsymbol{x}(1) \sim p(\boldsymbol{x})$ be a sample from the target distribution, and $t \sim \text{Unif}([0, 1])$, we define interpolated points as $\boldsymbol{x}(t) = (1 - t) \cdot \boldsymbol{x}(0) + t \cdot \boldsymbol{x}(1)$, and train the velocity field to minimize the squared error from the slope of the straight line connecting $\boldsymbol{x}(0)$ and $\boldsymbol{x}(1)$. After training the velocity field $v_\theta(t, \boldsymbol{x}(t))$, flow-matching runs numerical integration to compute $\psi_\theta(t, \boldsymbol{x}(0))$. This numerical integration procedure makes several calls to compute the velocity field.

## 4 FLOQ: TRAINING Q-FUNCTIONS WITH FLOW-MATCHING

In this section, we introduce the our proposed approach, `floq` (flow-matching Q-functions), which leverages iterative computation with dense supervision to train Q-functions. To do so, we address the two central questions needed to make `floq` work: **(a)** how to handle moving target values in the training loss for a flow-based Q-function and **(b)** how to do effective flow-matching over scalar Q-values without collapse for learning. Flow-matching preliminaries are discussed in Appendix A.1.

### 4.1 FLOQ PARAMETERIZATION

In contrast to standard deep Q-networks that map state-action pairs to scalar values, floq parameterizes a time-dependent, state-action-conditioned velocity field $v_\theta(t, \boldsymbol{z} \mid s, a)$ over a one-dimensional latent input $\boldsymbol{z} \in \mathbb{R}$. At $t = 0$, this input $\boldsymbol{z}$ is sampled from the uniform distribution $\mathrm{Unif}[l, u]$, where $l$ and $u$ are scalars that define the range of initial sample noise used for training. The velocity field transforms the initial sample $\boldsymbol{z}$ into a distribution over the Q-value. We will train floq such that the learned distribution of Q-values match a Dirac-Delta around the groundtruth Q-function, i.e., $\psi_\theta(1, \boldsymbol{z}|s, a) \sim \delta_{Q^\pi(s,a)}$ at $t = 1$. We can obtain the Q-value sample by numerically integrating the ODE using the Euler method. One instantiation is shown below. $\forall j \leq K$:

$$\psi_\theta(j/K, \boldsymbol{z} \mid s, a) = \boldsymbol{z} + \tfrac{1}{K} \sum_{i=1}^{j} v_\theta\left(\tfrac{i}{K}, \psi_\theta\left(i-1/K, \boldsymbol{z} \mid s, a\right) \Big| s, a\right), \; Q(s, a, \boldsymbol{z}) := \psi_\theta(1, \boldsymbol{z} \mid s, a) \quad (4.1)$$

An example illustration of this process is shown in Figure 1. This iterative process enables us to dynamically adjust the Q-function by varying the number of integration steps $K$, by controlling the number of evaluations of the velocity field $v_\theta$, and thereby the "depth" of the model. Finally, we remark that although Equation 4.1 may appear similar to performing averaging like an ensemble, it is fundamentally different: the inputs passed to the velocity field $v_\theta$ at each step $i$ depend on its own outputs from the previous step $i - 1$. This recursive dependence introduces a form of iterative computation that is absent in conventional ensembles, that perform computation in parallel. As we demonstrate in our experiments (Section 5), this formulation enjoys greater benefits of scale than simply ensembling independent neural networks without iterative computation. In practice, the velocity field $v_\theta(i/K, \cdot|s, a)$ can be conditioned on the various representations of the intermediate Q-values $\psi_\theta(i-1/K, \cdot|s, a)$ to improve the effectiveness of learning. We opt to use a categorical representation of $\psi_\theta$ when passing it as input to the velocity network. We discuss this in Section 4.3.

### 4.2 TRAINING LOSS FOR THE FLOQ ARCHITECTURE

With this parameterization in place, the next step is to design a training loss for the velocity field. Building upon TD-learning and flow-matching methods, a natural starting point is to iteratively train the velocity field using a loss that resembles linear flow-matching (Equation 4.2), but with targets obtained via Bellman bootstrapping. This is akin to TD-flows (Farebrother et al., 2025) and $\gamma$-models (Janner et al., 2020) that train a generative dynamics model with TD-bootstrapped targets. To do so, we introduce a target velocity field $\tilde{v}_\theta(t, \boldsymbol{z} \mid s, a)$, parameterized as a stale moving average of the main velocity field $v_\theta$, similar to target networks in standard value-based RL. Given a transition $(s, a, r, s')$, we first sample an action $a' \sim \pi(\cdot \mid s')$ from the current policy at the next state s', and compute target Q-value samples $\psi_{\tilde{\theta}}(1, \boldsymbol{z}' \mid s', a')$ by integrating the target flow, starting from some $\boldsymbol{z}'$ (via Euler integration) to obtain the predicted Q-value sample, $\psi_{\tilde{\theta}}(1, \boldsymbol{z}' \mid s', a')$.

We then average these predicted Q-value samples $\psi_{\tilde{\theta}}(1, \boldsymbol{z}' \mid s', a')$ for several values of the initial noise $\boldsymbol{z}'$ to compute an estimate of the target expected Q-value $Q_{\tilde{\theta}}(s', a')$. The bootstrapped TD-target is given by: $y(s, a) = r(s, a) + \gamma \frac{1}{m} \sum_{j=1}^{m} \psi_{\tilde{\theta}}(1, \boldsymbol{z}'_j \mid s', a')$, where $r(s, a)$ denotes the reward estimate for transition (note that this is distinct from distributional RL). We use this mean Q-target to train the Q-value at state-action pair $(s, a)$ by regressing to the target $y(s, a)$ via a linear flow-matching loss. Concretely, given a $t \sim \mathrm{Unif}[0, 1]$, we construct an **interpolant** between noise $\boldsymbol{z}$ sampled at the initial step and the target Q-value $y$, $\boldsymbol{z}(t) = (1 - t) \cdot \boldsymbol{z} + t \cdot y(s, a)$, and train the velocity at this interpolant to match the displacement from $\boldsymbol{z}(0)$ to $y$ via flow-matching (Equation 4.2):

$$\mathcal{L}_{\texttt{floq}}(\theta) = \mathbb{E}_{\boldsymbol{z}, t}\left[\left\|v_\theta(t, \boldsymbol{z}(t) \mid s, a) - \frac{(y(s, a) - \boldsymbol{z})}{1 - 0}\right\|_2^2\right]. \quad (4.2)$$

### 4.3 PREVENTING FLOW COLLAPSE: HOW TO MAKE FLOQ WORK WELL?

So far, we have introduced a conceptual recipe for parameterizing and training a Q-function critic via flow matching. However, a naïve instantiation of this idea performed no better than a standard monolithic Q-function in our initial experiments. This performance is the result of the inability of the network to meaningfully condition on the interpolant $\boldsymbol{z}(t)$, leading the flow model to often collapse to a monolithic Q-network (Figure 2). Interestingly, we find that this problem can is a result of two peculiarities associated with applying flow-matching to TD: training with constantly evolving targets and running the flow on a scalar Q-values. We describe our approach for handling these pathologies, and to do so, we first answer: *what constitutes a "healthy"* floq *velocity field?* Then we introduce two crucial modifications to the floq architecture that enable learning healthy floq networks.

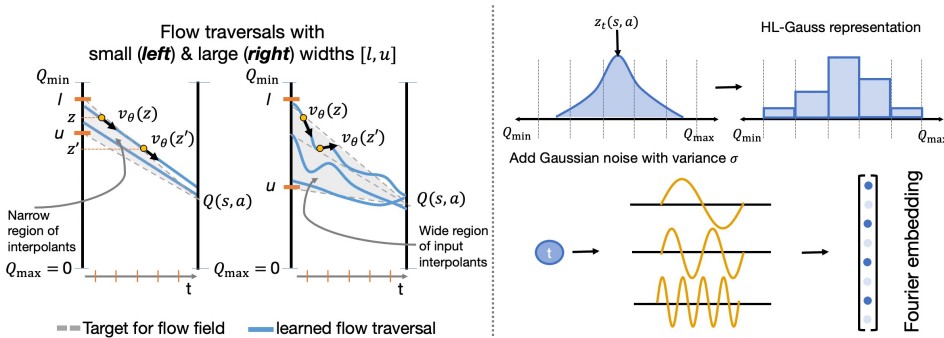

Figure 2: *Illustrating the role of our design choices.* **Left:** When the width of the interval $[l, u]$ is small, and the overlap between this interval and the range of target Q-values we hope to see is minimal, we would expect to see more straight flow traversals, that might be independent of interpolant $z$. However, with wider intervals $[l, u]$, the flow traversal would depend on $z$, and hence span a curved path when running numerical integration during inference. **Right:** Illustrating how we transform an input interpolant $z$ into a categorical representation (top) and converting time $t$ into a Fourier-basis embedding (bottom).

***When is* `floq` *effective?*** Unlike traditional applications of flows, `floq` applies them to scalar Q-values. How does flow matching on a scalar work? Consider the trajectory traced by the flow during inference, as it evolves from initial noise ($t = 0$) to the Q-value estimate produced by the network ($t = 1$) (Figure 2; left). If this trajectory is a straight line, the velocity field $v_\theta(z(t), t)$ does not need to depend on $t$ and predicting a constant velocity proportional to the target Q-value is sufficient. In this case, flow matching provides no additional capacity beyond a monolithic Q-network. In contrast, if the trajectory is curved, the velocity field must utilize the interpolant $z(t)$ and time $t$ to predict customized velocities and be able to integrate to an accurate Q-value estimate at $t = 1$. Thus, even though training uses a simple linear flow-matching loss, extra capacity emerges only when the learned flows produce (slightly) curved trajectories. Here, iterative computation amplifies model capacity, allowing `floq` to outperform monolithic Q-networks. Note that overly curved flows are also problematic as they amplify errors in the integration process itself. Therefore, we want to attain an intermediate sweet spot in regards to the straightness of the traversals (see Figure 13).

***Design choice 1: Distribution of the initial noise sample.*** As shown in prior works (Lipman et al., 2023; Liu et al., 2023), rescaling the source noise leaves the target distribution unchanged, but it alters the curvature of the transport trajectories. Interestingly, this effect seems to be particularly pronounced when applying flow-matching to scalar TD-learning (see Figure 13 in experiments). As such, we find that that setting the bounds $l$ and $u$ for the distribution of the initial noise, Unif $[l, u]$ greatly affects the performance of `floq`. We hypothesize that two aspects are important: (a) how close the target Q-values during training are to the chosen interval $[l, u]$, and (b) the width of the interval $u - l$. If the width $u - l$ is too small, then the interpolants $z(t)$ span only a very limited range of values. When we then run (imperfect) TD-loss training on these interpolants, the network parameterizing the velocity field receives little meaningful variation in $z(t)$ to associate changes in target values with. As a result, the model fails to exploit $z(t)$ effectively and degenerates into behaving like a standard monolithic Q-function. Likewise, if the interval $[l, u]$ is very disjoint from the range of target Q-values during training, then all interpolants $z(t)$ are forced to predict large velocities pointing in the general direction of the target Q-value. This reduces the need to learn calibrated velocity predictions conditioned on $z(t)$ and time $t$. We show this in Figure 2, left.

Thus, we propose to choose $l$ and $u$ using a simple heuristic. We set $u = Q_{\max}$ (=0 for most of our tasks). We then choose $l \geq Q_{\min}$ to maximize the interval width $u - l$ while yielding stable learning curves and TD-error values comparable to a monolithic network. Here $Q_{\min}$ and $Q_{\max}$ denote the minimal and maximal possible Q-value achievable on the task. Thus, the interval $[l, u]$ is likely to overlap well with the target Q-values the velocity field must predict during training. We remark that this heuristic relies on the assumption that over the course of learning, Q-value predictions will evolve from near-zero values to the target value. Since standard network initializations already produce values near zero, this assumption is not limiting and helps cover the target Q-value range we see.

***Design choice 2: Representing interpolant inputs to the velocity network.*** The second challenge arises because the magnitudes of the scalar interpolant $z(t)$ evolve during training. While standard TD-learning naturally handles non-stationary *outputs* (Q-values growing from near-zero random initialization), this is usually manageable with best practices such as activation normalization (Nauman et al., 2024). In contrast, `floq` must cope with non-stationarity at the *input*, since the interpolant $z(t)$

is fed into the velocity flow. As training progresses, its magnitude grows, leading to large gradients and activations in the network. To address this, we adopt a categorical representation of *input* $z(t)$ (output is still scalar like baselines), inspired by the HL-Gauss encoding of Farebrother et al. (2024), which prevents $z(t)$'s magnitude from skewing activations. Concretely, we add Gaussian noise with standard deviation $\sigma$, then convert the resulting PDF $\mathcal{N}(z(t), \sigma^2)$ into a categorical histogram over $N$ bins spanning the expected Q-value range. In contrast to Farebrother et al. (2024), we use a larger $\sigma$, such that roughly 80% of bins receive non-zero mass at initialization, encouraging broader coverage. Finally, we also utilized a Fourier basis representation of the time variable $t$, provided as input to the velocity network $v_\theta(t, z(t))$. This is illustrated in Figure 2 (right). We show in our experiments than doing so helps substantially by encouraging the network to meaningfully utilize this time.

> **Summary: `floq` architecture and training**
>
> `floq` parameterizes the Q-function via a learned velocity field, trained with a linear flow-matching loss against the target Q-value (Eq.4.2). To scale capacity, we sample initial noise $\mathrm{Unif}[l, u]$ broadly to overlap with target Q-values. The interpolant is encoded categorically, and the input $t$ to $v_\theta$ is Fourier-encoded. See Algorithm 1 for details.

**Additional implementation details.** We build on FQL, and use a similar approach for training the policy. This way, we isolate the performance differences to our proposed flow-matching critic. In addition, because `floq` parameterizes a flow-matching critic, extracting reparameterized policy gradients requires computing gradients through the full integration process with respect to the input action, which is costly. To alleviate this, we adapt a technique from Park et al. (2025d) for flow-matching policies and apply it to critics. Specifically, we train a *distilled critic*, $Q_\psi^{\mathrm{distill}}(s, a)$, to approximate the predictions obtained by integrating the flow critic, $Q_\theta^{\mathrm{flow}}(s, a, z)$. Policy extraction is then performed directly on the distilled critic. We illustrate this idea in Algorithm 1.

## 5 EXPERIMENTAL EVALUATION

The goal of our experiments is to evaluate the efficacy of `floq` in improving offline RL and online fine-tuning. To this end, we compare `floq` to state-of-the-art methods, and answer the following questions: **(1)** Does `floq` improve performance when compared to using similar-sized networks on benchmark tasks? and **(2)** How does the use of iterative computation via `floq` compare with the use of "parallel" computation of a neural network ensemble and "sequential" iterative computation driven by ResNets of comparable size? We then run several experiments to understand the behavior of `floq` critics. Furthermore, we run a variety of ablation studies to understand the design choices that drive the efficient use of iterative computation, including the roles of **a)** tuning the width of initial noise sample, **b)** categorical representations of the interpolant input, and **c)** Fourier-basis time embeddings.

### 5.1 MAIN OFFLINE RL RESULTS

**Offline RL tasks and datasets.** Following evaluation protocols from recent work in offline RL (Park et al., 2025d; Wagenmaker et al., 2025; Espinosa-Dice et al., 2025), we use the **OGBench** task suite (Park et al., 2025a) as our main evaluation benchmark (see Figure 18). OGBench provides a number of diverse, challenging tasks across robotic locomotion and manipulation, where these tasks are generally more challenging than standard D4RL tasks (Fu et al., 2020), which have been saturated as of 2024 (Tarasov et al., 2023; Rafailov et al., 2024; Park et al., 2024). While OGBench was originally designed for benchmarking offline goal-conditioned RL, we use its reward-based single-task variants ("`-singletask`" from Park et al. (2025d)). We employ 5 locomotion and 5 manipulation environments where each environment provides 5 tasks, totaling to **50** state-based OGBench tasks. Some tasks are more challenging and longer-horizon (e.g., marked in the table).

**Comparisons and evaluation protocol.** In addition to a `floq` critic, our experiments use flow-matching policies. Thus, flow-Q learning (**FQL**) (Park et al., 2025d), which utilizes a flow-matching policy with a monolithic Q-network, is our main comparison. FQL reports results with 1M training steps; we additionally re-run FQL for 2M steps and report both results. We run `floq` with a default set of hyperparameters across tasks, but on the more challenging `humanoidmaze-large` and `antmaze-giant` tasks we found a larger batch size of 512 to be more effective, which we use for both FQL and `floq`. Beyond FQL, we compare against three recent SOTA offline RL algorithms, all of which rely on monolithic Q-functions: (i) **ReBRAC** (Tarasov et al., 2023), the strongest-performing method with a monolithic Q-network and a Gaussian policy; (ii) **DSRL** (Wagenmaker et al., 2025)

Table 1: **Offline RL results (all tasks).** `floq` achieves competitive or superior performance compared to prior approaches. "Hard" environments refers to the set of environments where the FQL approach attains below 50% performance, averaged over the 5 tasks. `floq` is especially more performant on these hard environments over prior comparisons, where its performance (with best configuration) is around $1.8\times$ of FQL. We don't report DSRL (Wagenmaker et al., 2025) here as this prior work does not run on the exhaustive set of tasks (see Table 2 for these). A comparison on just the default tasks reveals `floq` outperforms DSRL by $> 2\times$.

| Env. (5 tasks each) | Gaussian Policy | | Flow Policy | | | Flow Q-function (Ours) | |
| --- | --- | --- | --- | --- | --- | --- | --- |
| | BC | ReBRAC | SORL | FQL (1M) | FQL(2M) | floq (Def.) | floq (Best) |
| antmaze-large | $11_{\pm1}$ | $81_{\pm5}$ | $89_{\pm2}$ | $79_{\pm3}$ | $83_{\pm5}$ | $\mathbf{91}_{\pm5}$ | $\mathbf{91}_{\pm5}$ |
| antmaze-giant *(Hard)* | $0_{\pm0}$ | $26_{\pm8}$ | $9_{\pm6}$ | $22_{\pm19}$ | $27_{\pm23}$ | $36_{\pm21}$ | $51_{\pm12}$ |
| hmmaze-medium | $2_{\pm1}$ | $22_{\pm8}$ | $64_{\pm4}$ | $57_{\pm5}$ | $69_{\pm20}$ | $\mathbf{82}_{\pm10}$ | $\mathbf{82}_{\pm10}$ |
| hmmaze-large *(Hard)* | $1_{\pm0}$ | $2_{\pm1}$ | $5_{\pm2}$ | $9_{\pm6}$ | $16_{\pm9}$ | $\mathbf{28}_{\pm9}$ | $\mathbf{28}_{\pm9}$ |
| antsoccer-arena | $1_{\pm0}$ | $0_{\pm0}$ | $\mathbf{69}_{\pm2}$ | $60_{\pm2}$ | $61_{\pm10}$ | $65_{\pm12}$ | $65_{\pm12}$ |
| cube-single | $5_{\pm1}$ | $91_{\pm2}$ | $97_{\pm1}$ | $96_{\pm1}$ | $94_{\pm5}$ | $\mathbf{98}_{\pm3}$ | $\mathbf{98}_{\pm3}$ |
| cube-double *(Hard)* | $2_{\pm1}$ | $12_{\pm1}$ | $25_{\pm3}$ | $29_{\pm2}$ | $25_{\pm6}$ | $\mathbf{47}_{\pm15}$ | $\mathbf{47}_{\pm15}$ |
| scene | $5_{\pm1}$ | $41_{\pm3}$ | $57_{\pm2}$ | $56_{\pm2}$ | $57_{\pm4}$ | $\mathbf{58}_{\pm6}$ | $\mathbf{58}_{\pm6}$ |
| puzzle-3x3 *(Hard)* | $2_{\pm0}$ | $21_{\pm1}$ | $-_{\pm-}$ | $30_{\pm1}$ | $29_{\pm5}$ | $\mathbf{37}_{\pm7}$ | $\mathbf{37}_{\pm7}$ |
| puzzle-4x4 *(Hard)* | $0_{\pm0}$ | $14_{\pm1}$ | $-_{\pm-}$ | $17_{\pm2}$ | $9_{\pm3}$ | $21_{\pm5}$ | $\mathbf{28}_{\pm6}$ |
| Avg Score (All Envs.) | 3 | 31 | $-$ | 46 | 47 | 56 | **59** |
| Avg Score (Hard Envs.) | 1 | 15 | $-$ | 21 | 21 | 34 | **38** |

that adapts a diffusion-based behavior cloning policy by performing RL over its latent noise space, improving over FQL; and (iii) **SORL** (Espinosa-Dice et al., 2025) that leverages shortcut flow models to improve upon FQL. Note that none of them utilize a flow-matching Q-function, but do innovate across various properties of policy training. Comparing to the strongest methods that innovate on the policy allows us to evaluate the importance of a flow-matching Q-function.

**floq configuration.** We run `floq` in two configurations. The "default" configuration utilizes the same hyperparameters across tasks, whereas the "best" configuration uses environment-specific hyperparameters that still are fixed across all tasks in that environment, to get a sense of the upper bound with `floq`. Even the default configuration of `floq` substantially outperforms all prior methods. The default configuration utilizes $K = 8$ flow steps and sets the width of $u - l$ to be $\kappa \times (Q_{\max} - Q_{\min})$, where $\kappa = 0.1$. The best configuration tunes the number of flow steps $K \in \{4, 8\}$ and $\kappa \in \{0.1, 0.25\}$ per environment (*not* per task). For a fair comparison with FQL, we use a 4-layer flow critic in all environments, except in the cube (single and double) environments where we employ a smaller 2-layer flow critic because we saw training instabilities with 4-layer critics.

**Empirical results.** Observe in Table 1 that `floq` outperforms prior methods, including FQL, on average across all 50 tasks, evaluated over 3 seeds for each task. Also note that `floq` improves over FQL most on the harder environments, where FQL attains performance below 50% success rate (antmaze-giant, hmmaze-large, cube-double, puzzle-3x3, and puzzle-4x4). For a statistically rigorous evaluation, we adopt techniques from Agarwal et al. (2021) and plot various statistics of the comparisons between `floq` and FQL: **1)** median and IQM scores in Figure 8, where we do not observe *any* overlap between the confidence intervals; **2)** performance profile and $P(X > Y)$ statistic in Figure 9, which are both strictly in favor of `floq`. Since DSRL (Wagenmaker et al., 2025) only evaluates on the 10 default OGBench tasks, we also provide an additional results table on only the default tasks in each environment in the appendix (Table 2). On the default tasks, we find that `floq` outperforms DSRL (20% for DSRL vs. 45% for `floq`), improving by over $2\times$ in success rate. These results establish the efficacy of `floq`. While `floq` uses an expected Q-value backup it still learns a stochastic Q-function. Thus, we also compare `floq` to a representative distributional RL approach (IQN (Dabney et al., 2017)) in Table 2, and we observe that `floq` outperforms IQN.

## 5.2 MAIN ONLINE FINE-TUNING RESULTS

Next, we evaluate `floq` in the online RL fine-tuning setting. Here, we first train agents completely offline for $1M$ steps and subsequently fine-tune them online for an additional $2M$ steps (Figure 3). Across four challenging tasks (humanoidmaze-medium, humanoidmaze-large, antmaze-giant, and antsoccer-arena), `floq` provides a substantially stronger initialization, faster learning during online interaction, and converges to higher final performance than FQL. Our complete set of results (Figure 10), show a similar trend establishing the efficacy of `floq` in online fine-tuning.

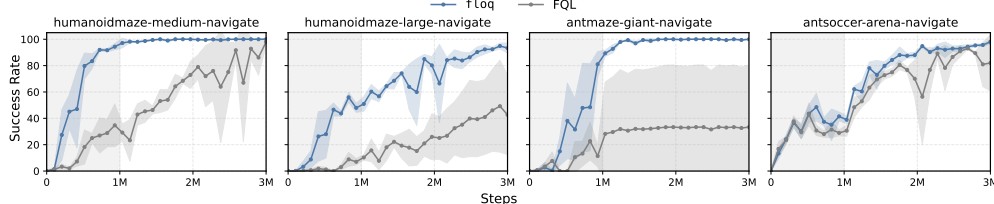

Figure 3: ***Learning curves for online fine-tuning*** of `floq` and FQL. `floq` not only provides a stronger initialization from offline RL training but also maintains its advantage throughout online fine-tuning on the hardest tasks, leading to faster adaptation, and higher final performance. The shaded area denotes offline RL.

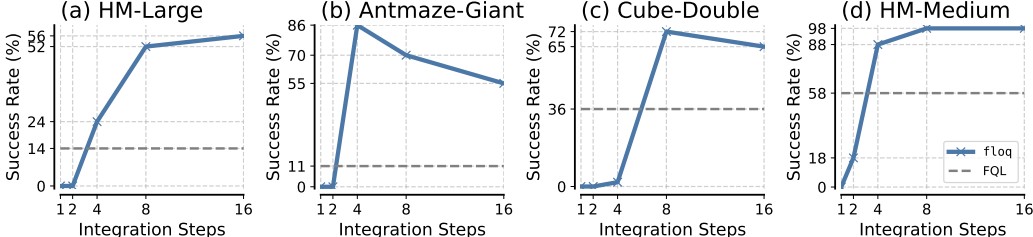

Figure 4: ***Effect of integration steps on `floq`.*** Performance of `floq` with varying flow steps, compared against a monolithic Q-function (FQL). More flow steps generally improve performance, but too many steps can lead to diminishing or negative returns (e.g., `antmaze-giant`). That said, in all configurations, `floq` outperforms FQL, and utilizing a moderately large number of flow steps is important.

## 5.3 UNDERSTANDING THE SCALING PROPERTIES AND BEHAVIOR OF `FLOQ`

To better understand the benefits of iterative compute in `floq`, we analyze its scaling behavior and compare it to different approaches. Specifically, we study: **(i)** how the number of flow-integration steps controls the expressivity of `floq`; **(ii)** how `floq`'s iterative computation compares to monolithic critic scaling; and **(iii)** the importance of applying supervision to the velocity field at every flow step.

***1) How does the performance of `floq` depend on the number of integration steps for the flow?***
We now study the effect of varying the number of integration steps for the flow in `floq`. In Figure 4, we report the success rate of `floq` with $K \in \{1, 2, 4, 8, 16\}$ flow steps, alongside a monolithic Q-function (FQL) using the same architecture. Increasing the number of flow steps generally improves the performance of `floq`, with notable gains on harder tasks, `hmmaze-large` and `antmaze-giant`. Importantly, even with just 4 flow steps, `floq` already outperforms FQL, and the gap widens further with additional steps. However, we also observe diminishing returns and, in some cases, slight degradation beyond a moderate number of steps (e.g., on `antmaze-giant`, where 8 and 16 steps perform worse than 4). We suspect that this degradation stems from overfitting when the number of integration steps for computing the target is excessive. A similar degradation is observed for ResNets in Figure 6. We discuss this further in Appendix A.11.

***2) How does `floq` compare against increasing "sequential" compute of monolithic critics?*** One hypothesis is that `floq` could be implementing a similar iterative computation strategy such as ResNet (He et al., 2016). Unlike `floq`, ResNet does not utilize dense supervision after each computation block. To test whether this matters, we compare `floq` to a ResNet. We use the same 4-layer flow critic from before and benchmark it against the best-performing ResNets with FQL. We ran a cross-product over possible ResNet configurations with block sizes of $2, 4, 8, 16$ layers and blocks of $8, 4, 2$. These configurations cover all ways to build a ResNet where 32 layers are involved in a forward pass.

We compare `floq` to the best ResNet configuration under a fixed total inference compute budget (that is, an upper bound on the number of feed-forward layers) on humanoidmaze-large and antmaze-giant, since these harder environments should benefit from bigger ResNets. As shown in Figure 6, while ResNet critics do improve over FQL, they remain worse than `floq` even under matched inference compute. Note that `floq` does not itself utilize a ResNet, though its velocity network could use residual layers. Also note that while ResNet architectures instantiate a new set of parameters for every layer added to the network, `floq` uses parameters from a single 4-layer MLP for all values of inference capacity. This shows that the gains of flow critics are not due to adding more residual

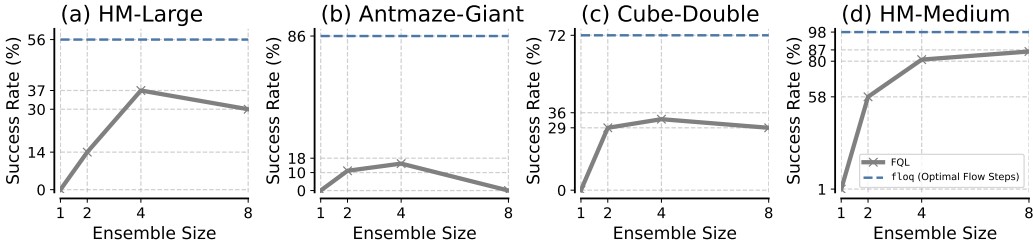

Figure 5: **Comparison of `floq` with monolithic ensembles.** We evaluate ensembles of size 1, 2, 4, and 8 of FQL critics. While larger ensembles do better, even an 8-critic ensemble falls short of `floq` with the same number of flow steps, showing that flow critics provide gains beyond parallel compute.

layers or parameters, but rather the dense supervision provided by supervising the velocity field at each step of iterative computation.

*3) How does `floq` compare to scaling monolithic critics with ensembles?* A common way to expand Q-function capacity is through ensembling, averaging predictions from multiple critics for backups and policy updates. This increases parallel rather than sequential compute, and is attractive if effective. We trained ensembles of 1, 2, 4, and 8 critics (each the same size as the base FQL critic) and compared them to `floq`. As shown in Figure 5, ensembles yield only modest gains over FQL, and even 8 critics fail to match `floq`. Notably, `floq` also uses 8 forward passes (via integration steps), so the compute is comparable. Thus, the benefits of flow critics stem not from parallel averaging.

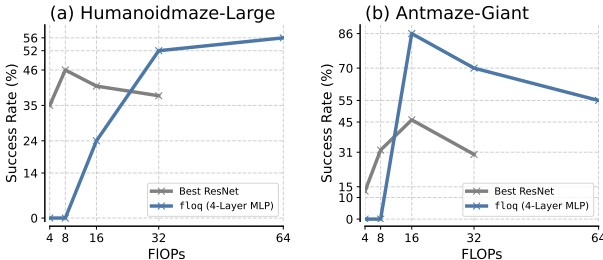

Figure 6: *Comparison of `floq` with ResNet critics on the hardest tasks*: `hmmaze-large`, `antmaze-giant`. A 4-layer flow critic outperforms the best ResNets under a total budget on forward pass capacity, even after tuning over multiple residual configurations. For any given value on the x-axis, we plot the performance of the best performing ResNet configuration at that inference cost. For `floq`, we run more integration steps at inference. Thus, our approach of training `floq` does not simply add more depth.

*4) Does `floq` benefit from densely supervising the velocity at all time steps?* To answer this question,

We trained a variant of `floq` where the velocity field was supervised only at $t = 0$ with a single flow step. While one might think that training `floq` at $t = 0$ is equivalent to baseline FQL, this is not the case. Even with one integration step, the input to the velocity network is still a scalar noise. The velocity field is trained to predict the difference between the target Q-value and this noise for all noise values, creating several auxiliary tasks rather than the single task in baseline FQL (noise set to 0). We hypothesize

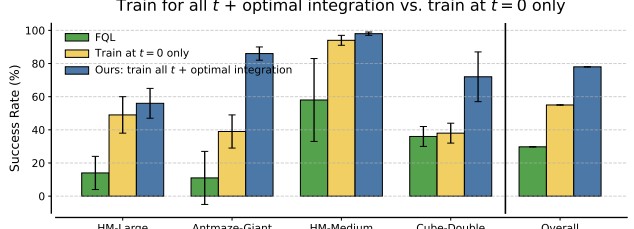

Figure 7: *Comparing FQL, training `floq` only at $t = 0$, and full `floq`* with supervision across all $t$ (and optimal integration steps $k$ chosen from $\{4, 8, 16\}$). While $t = 0$ training improves over FQL, full `floq` consistently achieves the best performance, showing the benefits from training at all integration steps.

that fitting these auxiliary tasks yields representational benefits, consistent with prior observations linking auxiliary losses with improved TD representations (Lyle et al., 2021). As shown in Figure 7, this restricted variant outperforms FQL but still underperforms `floq`, which supervises velocity at all $t \in [0, 1]$ and uses multiple steps. On humanoidmaze-large, performance improves from 14% (FQL) to 49% with only $t = 0$ training, while full `floq` reaches 56%. On antmaze-giant, the pattern is stronger, with scores of 11%, 39%, and 86%. Similar trends hold on hm-medium (58%, 94%, 98%) and cube-double (36%, 38%, 72%). The smaller gap on hm-medium likely reflects its lower difficulty. Thus, while $t = 0$ already provides meaningful gains, supervising across all $t$ and using multiple steps is important for realizing the full potential of `floq`.

> **Takeaways: properties and behavior of `floq`**
>
> More integration steps help, but performance saturates and can degrade at very high values. `floq` outperforms approaches that scale either parallel or sequential compute for monolithic Q-functions, and multiple steps are needed for best performance.

### 5.4 ABLATION STUDIES FOR FLOQ

Finally, in Appendix A.5, we present experiments ablating various design choices and hyperparameters. We evaluate the sensitivity of `floq` to these choices and give thumb rules for tuning them. The choices are: **a)** the range $[l, u]$, for the support for the initial noise sample $z(0)$, **b)** the approach for embedding the flow step "time" $t$, and **c)** the approach for embedding the interpolant $z(t)$.

> **Takeaways: ablation studies for `floq`**
>
> 1) Utilizing an HL-Gauss embedding for $z(t)$ is crucial (Figure 14). 2) Utilizing a Fourier-basis embedding of time is critical (Figure 12). 3) A moderate width of the initial noise distribution improves flow curvature, and performs best (Figure 13).

## 6 DISCUSSION AND PERSPECTIVES ON FUTURE WORK

In this paper, we presented `floq`, an approach for training critics in RL using flow-matching. `floq` formulates value learning as transforming noise into the value function via integration of a learned velocity field. This formulation enables scaling Q-function capacity by utilizing more compute during the process of integration to compute the Q-function. As a result of utilizing a flow-matching objective for training, `floq` utilizes dense supervision at every step of the integration process. We describe some important design choices to train flow-matching critics to make meaningful use of integration steps. Through our experiments, we show that `floq` attains state-of-the-art results on a suite of commonly-used offline RL tasks, and outperforms other ways of expanding capacity of a Q-function (e.g., via a ResNet or monolithic Q-function ensemble). We also show the necessity of learning curved flow traversals to make effective use of capacity and utilizing the design choices we prescribe in this work.

**Future work.** We believe `floq` presents an exciting approach to scale Q-function capacity. Thus, there are a number of both theoretical and empirical open questions. From an empirical standpoint, it is important to understand how to appropriately set the number of integration steps as excessive steps may degrade Q-function quality. This degradation, however, is not localized to just flows but also to ResNets (Figure 6), indicating that this is perhaps a bigger issue with TD-learning. Another interesting direction is to build new methods and workflows for using Q-functions that rely on the property that `floq` inherently represents a "cascaded" family of critics with different capacities—all within one network. Can this property be used for tuning network size upon deployment, cross-validation of model size, or improving efficiency of policy extraction? Answering this question would be interesting for future work. Finally, `floq` also provides one possibility for sequential or "depth"-based test-time scaling for value functions. Studying how this sort of sequential scaling can be combined with parallel scaling (i.e., ensembles) and horizon reduction techniques (Park et al., 2025b) would be interesting as well.

From a theoretical standpoint, quantifying iterative computation properties of `floq` would be impactful: in principle, curved flow traversals should enable the critic network to spend more test-time compute (i.e., integration steps) to perform equivalents of "error correction" and "backtracking" from large language models (LLMs) (Guo et al., 2025), but now in the space of scalar, continuous values to better approximate the target Q-function. We believe formalizing this aspect would not only be impactful for value-based RL, but could also shed light on methods to use test-time compute in flow/diffusion models in other domains. Second, our results show that there are substantial representation learning benefits of `floq`. We believe that studying the mechanisms and differences between feature learning induced by `floq` compared to standard TD-learning with regression (Kumar et al., 2022) or classification (Farebrother et al., 2024) would be interesting for future value-based methods. `floq` also provides a rich family of auxiliary tasks to train a critic, which provides another angle to explain and study its properties. All of these are impactful directions to study in future work.

## ACKNOWLEDGEMENTS

We thank Max Sobol Mark, Zheyuan Hu, Aravind Venugopal, Seohong Park, Mitsuhiko Nakamoto, and Amrith Setlur for feedback on an earlier version of the paper and informative discussions. We thank members of the CMU AIRe lab for feedback and support. AK thanks Max Simchowitz for discussions on flow-matching and diffusion models. This work was supported by a Schmidt Sciences AI2050 fellowship and the Office of Naval Research under N00014-24-12206. We thank the TRC program of Google Cloud and NCSA Delta for providing computational resources that supported this work. This work used the Delta advanced computing and data resource at the National Center for Supercomputing Applications (NCSA) through allocation CIS250548 from the Advanced Cyberinfrastructure Coordination Ecosystem: Services And Support (ACCESS) program. Delta is supported by the National Science Foundation (award OAC-2005572) and the State of Illinois, and ACCESS is supported by U.S. National Science Foundation grants 2138259, 2138286, 2138307, 2137603, and 2138296.

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

# A APPENDICES

## A.1 OFFLINE RL PRELIMINARIES

We operate in the offline RL (Levine et al., 2020) problem setting, where the replay buffer $\mathcal{P}$ corresponds to a static dataset of transitions $\mathcal{D} = \{(s, a, r, s')\}$ collected using a behavior policy $\pi_\beta$. Our goal in this setting is to train a good policy using the offline dataset $\mathcal{D}$ alone. The Q-network, $Q_\theta$ is typically parameterized by a deep network (e.g., an MLP).

**Offline RL algorithms.** Offline RL methods aim to learn a policy that maximizes reward while penalizing deviation from the behavior policy $\pi_\beta$, in order to mitigate the challenge of distributional shift. This objective has been instantiated in various ways, including behavioral regularization (Wu et al., 2019; Fujimoto & Gu, 2021; Tarasov et al., 2023; Park et al., 2025d; Kumar et al., 2019), pessimistic value function regularization (Kumar et al., 2020), implicit policy constraints (Peters & Schaal, 2007; Peng et al., 2019; Wang et al., 2020; Mark et al., 2024), and in-sample maximization (Kostrikov et al.; Xu et al., 2023; Garg et al., 2023). While our proposed `floq` architecture for Q-function parameterization is agnostic to the choice of offline RL algorithm, we instantiate it on top of FQL (Park et al., 2025d) that utilizes a flow-matching policy to better model multimodal action distributions. FQL trains the Q-function using the standard temporal-difference (TD) error from soft actor-critic (SAC) (Haarnoja et al., 2018), shown in Equation 3.1, and optimizes the policy to stay close to a behavior policy estimated via flow-matching.

## A.2 ADDITIONAL RESULTS FOR `FLOQ`

In this section, we provide some additional and complete results supplementing the ones in main paper.

1. Figure 8 presents median and IQM scores and Figure 9 presents performance profiles and $P(X > Y)$ statistic comparing `floq` with FQL on all the 50 tasks studied in the paper.

2. Figure 10 presents results for online fine-tuning on all 10 default tasks.

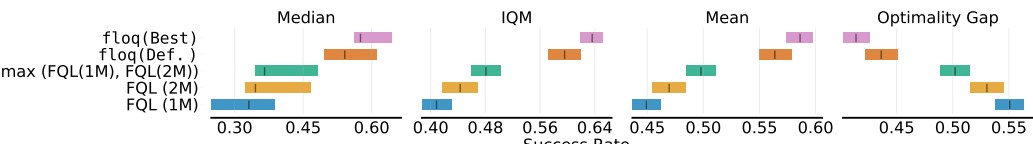

Figure 8: Comparison of `floq` against the baseline FQL across median, interquartile mean (IQM), mean and optimality gap, following Agarwal et al. (2021). Results show that `floq` consistently outperform FQL across all evaluation criteria with no confidence interval overlap in all cases, meaning that the gains from `floq` are significant.

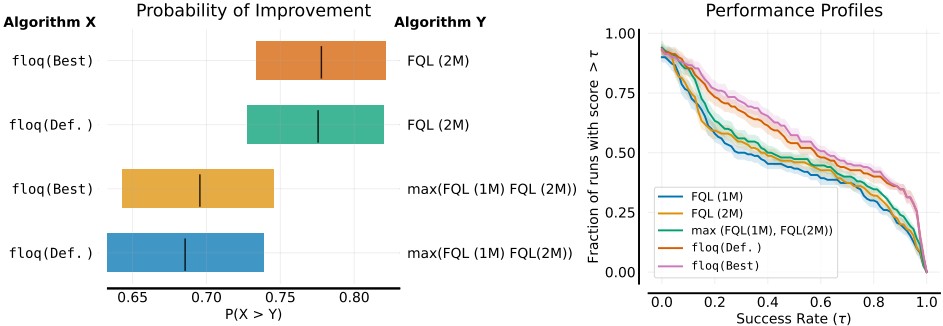

Figure 9: Comparison of `floq` against baseline FQL, following Agarwal et al. (2021). **Left:** Probability of Improvement $P(X > Y)$ showing that `floq` consistently outperform FQL across OGBench tasks. **Right:** Performance profiles illustrating that `floq` achieves higher scores across a larger fraction of runs compared to FQL.

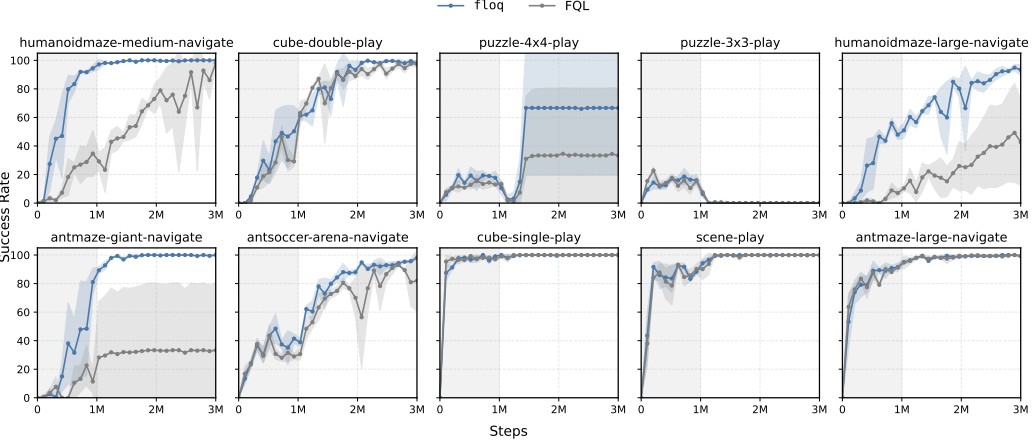

Figure 10: ***Learning curves for online fine-tuning*** of `floq` and FQL across all default tasks. `floq` not only provides a stronger initialization from offline RL training but also maintains its advantage through online fine-tuning, leading to faster adaptation and higher final success rates. The shaded gray area denotes offline RL training.

Table 2: **Offline RL results (Default Tasks).** floq achieves competitive or superior performance compared to the baselines. "Hard" tasks refers to the set of default tasks where the FQL baseline score is below 50% performance. floq is especially more performant on these hard tasks, more than doubling FQL's baseline performance.

| Env | Gaussian Policy | | Diff. Policy | Flow Policy | | | | | Flow Q-function (Ours) | |
| | BC | ReBRAC | DSRL | SORL | IQN | C51 | FQL (1M) | FQL (2M) | floq (Def.) | floq (Best) |
|---|---|---|---|---|---|---|---|---|---|---|
| antmaze-large | $0_{\pm0}$ | $91_{\pm10}$ | $40_{\pm29}$ | $93_{\pm2}$ | $86_{\pm1}$ | $79_{\pm3}$ | $80_{\pm8}$ | $85_{\pm4}$ | $\mathbf{94}_{\pm4}$ | $\mathbf{94}_{\pm4}$ |
| antmaze-giant | $0_{\pm0}$ | $27_{\pm22}$ | $0_{\pm0}$ | $12_{\pm6}$ | $5_{\pm6}$ | $0_{\pm0}$ | $11_{\pm16}$ | $14_{\pm29}$ | $70_{\pm8}$ | $\mathbf{86}_{\pm4}$ |
| hmmaze-medium | $1_{\pm0}$ | $16_{\pm9}$ | $34_{\pm20}$ | $67_{\pm4}$ | $27_{\pm17}$ | $23_{\pm16}$ | $19_{\pm12}$ | $58_{\pm25}$ | $\mathbf{98}_{\pm1}$ | $\mathbf{98}_{\pm1}$ |
| hmmaze-large | $0_{\pm0}$ | $2_{\pm1}$ | $10_{\pm12}$ | $20_{\pm9}$ | $22_{\pm5}$ | $6_{\pm2}$ | $8_{\pm5}$ | $14_{\pm10}$ | $\mathbf{52}_{\pm8}$ | $\mathbf{52}_{\pm8}$ |
| antsoccer-arena | $1_{\pm0}$ | $0_{\pm0}$ | $28_{\pm0}$ | $\mathbf{54}_{\pm5}$ | $44_{\pm3}$ | $36_{\pm7}$ | $39_{\pm6}$ | $49_{\pm11}$ | $49_{\pm10}$ | $49_{\pm10}$ |
| cube-single | $3_{\pm1}$ | $92_{\pm4}$ | $93_{\pm14}$ | $99_{\pm0}$ | $98_{\pm1}$ | $98_{\pm1}$ | $96_{\pm1}$ | $94_{\pm5}$ | $\mathbf{99}_{\pm2}$ | $\mathbf{99}_{\pm2}$ |
| cube-double | $0_{\pm0}$ | $7_{\pm3}$ | $53_{\pm14}$ | $33_{\pm8}$ | $57_{\pm2}$ | $40_{\pm11}$ | $36_{\pm6}$ | $29_{\pm8}$ | $\mathbf{72}_{\pm15}$ | $\mathbf{72}_{\pm15}$ |
| scene | $1_{\pm1}$ | $50_{\pm13}$ | $88_{\pm9}$ | $\mathbf{89}_{\pm9}$ | $80_{\pm4}$ | $82_{\pm11}$ | $76_{\pm9}$ | $78_{\pm7}$ | $83_{\pm10}$ | $83_{\pm10}$ |
| puzzle-3x3 | $1_{\pm1}$ | $2_{\pm1}$ | $0_{\pm0}$ | – | $20_{\pm3}$ | $13_{\pm4}$ | $16_{\pm5}$ | $14_{\pm4}$ | $17_{\pm6}$ | $17_{\pm6}$ |
| puzzle-4x4 | $0_{\pm0}$ | $10_{\pm3}$ | $\mathbf{37}_{\pm13}$ | – | $16_{\pm1}$ | $16_{\pm2}$ | $11_{\pm3}$ | $5_{\pm2}$ | $12_{\pm4}$ | $19_{\pm5}$ |
| Average Score (All) | 1 | 30 | 38 | – | 45 | 39 | 40 | 44 | 64 | **66** |
| Average Score (Hard) | 0 | 8 | 21 | – | 22 | 21 | 20 | 21 | 45 | **50** |

Table 3: *FQL ResNet performace on* `humanoidmaze-large` *and* `antmaze-giant`. $(m, n)$ indicates $n$ blocks each of depth $m$. For each fixed number of FLOPs $m \times n$, the best-performing architecture per environment is in bold.

| FLOPs $(m \times n)$ | HM-Large | Antmaze-Giant |
|---|---|---|
| 4 | $\mathbf{35} \pm 19$ (2,2) | $\mathbf{13} \pm 12$ (2,2) |
| | $14 \pm 10$ (4,1) | $11 \pm 16$ (4,1) |
| 8 | $\mathbf{46} \pm 11$ (2,4) | $31 \pm 13$ (2,4) |
| | $21 \pm 11$ (4,2) | $22 \pm 9$ (4,2) |
| | $22 \pm 11$ (8,1) | $\mathbf{32} \pm 14$ (8,1) |
| 16 | $\mathbf{41} \pm 7$ (2,8) | $32 \pm 13$ (2,8) |
| | $34 \pm 8$ (4,4) | $17 \pm 8$ (4,4) |
| | $26 \pm 9$ (8,2) | $\mathbf{46} \pm 11$ (8,2) |
| | $24 \pm 22$ (16,1) | $0 \pm 0$ (16,1) |
| 32 | $25 \pm 10$ (2,16) | $23 \pm 11$ (2,16) |
| | $28 \pm 13$ (4,8) | $18 \pm 14$ (4,8) |
| | $\mathbf{38} \pm 19$ (8,4) | $\mathbf{30} \pm 9$ (8,4) |
| | $0 \pm 0$ (16,2) | $0 \pm 0$ (16,2) |

### A.3 COMPATIBILITY OF FLOQ WITH DIVERSE POLICY CLASSES

One natural question is whether the benefits of floq arise specifically from pairing it with a flow-based policy, or whether the FloQ critic alone provides value when combined with standard policy architectures. To study this, we evaluate all combinations in the grid:

$$(\text{Monolithic critic vs. floq critic}) \times (\text{Gaussian policy vs. Flow policy}).$$

In particular, we add the missing configuration: **floq critic + Gaussian policy**.

To isolate the effect of the critic, we integrate the floq critic into the ReBRAC framework, which employs a Gaussian policy. The policy architecture and actor update rules are unchanged; only the Q-function parameterization is replaced with the floq critic.

Table 4 summarizes the results across the 10 default OGBench tasks. The floq critic yields substantial improvements over ReBRAC, increasing the average score from $30 \rightarrow 41$. Gains are particularly large on sparse-reward, long-horizon environment such as ANTMAZE-GIANT ($27 \rightarrow 91$) and HMMAZE-MEDIUM ($16 \rightarrow 57$). These improvements demonstrate that the floq critic offers benefits even when the policy is Gaussian.

Combined with the results in the main text (floq with FQL), these findings show that the floq critic is modular and compatible with diverse policy classes. Its advantages are not tied to flow-based policies, but instead arise from its parameterization of the $Q$ value.

Table 4: **FloQ critic is compatible with Gaussian policies.** Performance on the 10 default OGBench tasks when integrating the FloQ critic into the ReBRAC framework (Gaussian policy). The FloQ critic improves ReBRAC significantly, showing that FloQ's benefits are not tied to the flow policy used in FQL.

| Environment (Default Task) | ReBRAC | FloQ (ReBRAC) | FQL | FloQ (FQL) |
|---|---|---|---|---|
| antmaze-large | 95 ±4 | **97** ±3 | 85 ±4 | 94 ±4 |
| antmaze-giant | 27 ±22 | **91** ±6 | 14 ±20 | 70 ±8 |
| hmmaze-medium | 16 ±9 | 57 ±12 | 58 ±25 | **98** ±1 |
| hmmaze-large | 2 ±1 | 0 ±1 | 14 ±10 | **52** ±8 |
| antsoccer-arena | 0 ±0 | 1 ±2 | **49** ±11 | 49 ±10 |
| cube-single | 92 ±4 | 89 ±4 | 94 ±5 | **99** ±2 |
| cube-double | 7 ±3 | 3 ±3 | 36 ±6 | **72** ±15 |
| scene | 50 ±13 | 58 ±8 | 78 ±7 | **83** ±10 |
| puzzle-3x3 | 2 ±1 | 9 ±6 | 16 ±5 | **17** ±6 |
| puzzle-4x4 | 10 ±3 | 6 ±4 | 11 ±3 | **12** ±4 |
| Average Score | 30 | 41 | 45 | **64** |

Table 5: **Offline RL Results on D4RL Tasks.** `floq` performs similarly to the FQL baseline on the D4RL benchmarks, showing its robustness. Values show normalized returns (mean ± std).

| Environment | FQL | FloQ |
|---|---|---|
| antmaze-large-diverse-v2 | 85 ±5 | 82 ±5 |
| antmaze-large-play-v2 | 82 ±8 | 75 ±8 |
| antmaze-medium-diverse-v2 | 73 ±6 | 71 ±10 |
| antmaze-medium-play-v2 | 76 ±7 | 80 ±7 |
| antmaze-umaze-diverse-v2 | 85 ±7 | 78 ±17 |
| antmaze-umaze-v2 | 96 ±2 | 97 ±2 |
| pen-human-v1 | 51 ±10 | 58 ±9 |
| pen-cloned-v1 | 78 ±8 | 72 ±5 |
| pen-expert-v1 | 140 ±5 | 140 ±8 |
| door-human-v1 | 0 ±0 | 0 ±0 |
| door-cloned-v1 | 3 ±1 | 3 ±2 |
| door-expert-v1 | 104 ±0 | 104 ±0 |
| hammer-human-v1 | 1 ±1 | 1 ±1 |
| hammer-cloned-v1 | 15 ±13 | 10 ±8 |
| hammer-expert-v1 | 125 ±3 | 125 ±2 |
| relocate-human-v1 | 0 ±0 | 0 ±0 |
| relocate-cloned-v1 | 0 ±0 | 0 ±0 |
| relocate-expert-v1 | 106 ±2 | 108 ±2 |

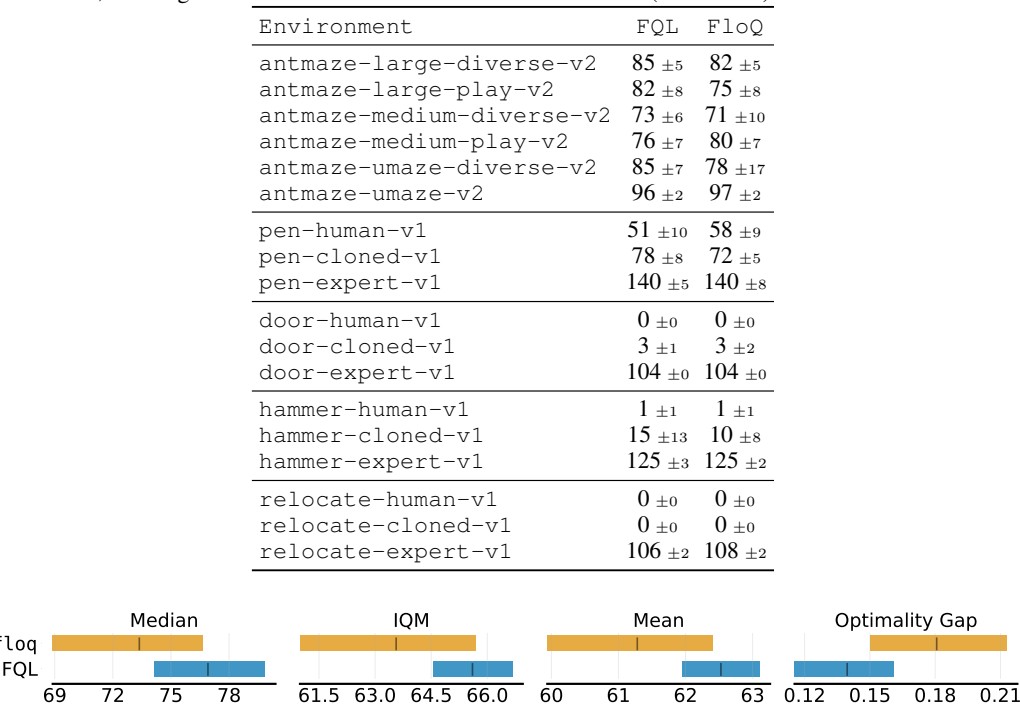

Figure 11: **Aggregate Performance on D4RL Tasks.** Comparison of `floq` and FQL on D4RL using Median, IQM, Mean, and Optimality Gap metrics shows that they perform similarly.

## A.4 RESULTS ON THE D4RL BENCHMARK

To further evaluate the robustness of `floq`, we also compare its performance to FQL on the standard D4RL benchmark. As shown in Table 5, `floq` performs similarly to the FQL baseline across all D4RL tasks, with nearly identical normalized returns on average. These tasks are considerably simpler than the long-horizon, sparse-reward OGBench environments used in the main evaluation, and the comparable results on D4RL demonstrate that `floq` does not sacrifice performance on easier benchmarks. In addition to per-task results, Figure 11 presents aggregate metrics using the rliable framework, including Median, IQM, Mean, and Optimality Gap. Across all four metrics, `floq` and FQL exhibit closely aligned confidence intervals, further indicating that their performance on D4RL is statistically indistinguishable.

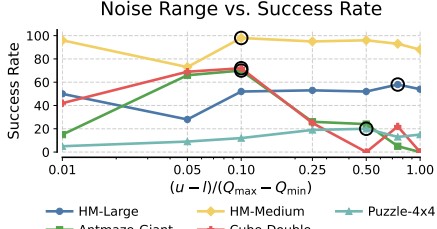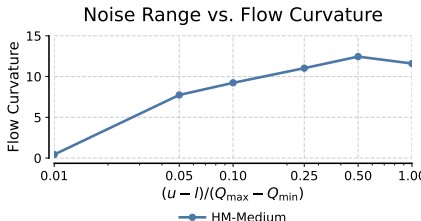

Figure 13: *Effect of variance of the initial noise sampling distribution on* `floq`. **Left:** Success rates across environments as a function of the initial noise scaling factor (black circles denote the best setting per environment). **Right:** Flow curvature in HM-Medium increases with noise variance, highlighting the tradeoff between too little curvature (flow collapses to monolithic critic) and too much curvature (difficult numerical integration).

Together, the table and aggregate statistics confirm that `floq` is robust across both challenging and standard offline RL benchmarks, matching FQL on D4RL while significantly outperforming it on the more challenging OGBench tasks.

## A.5 ADDITIONAL ABLATION STUDIES FOR `FLOQ`

**1) Ablations for the width of the $[l, u]$ interval.** We study the effect of varying the variance of the initial noise sample used in critic flow matching by expanding the width $u - l$ of the interval that the initial noise is sampled from. We present the results in Figure 13. On the left, we observe that the performance across several tasks typically peaks at intermediate variance values (note that the black circles marking the setting that yields the best success rate for each environ-

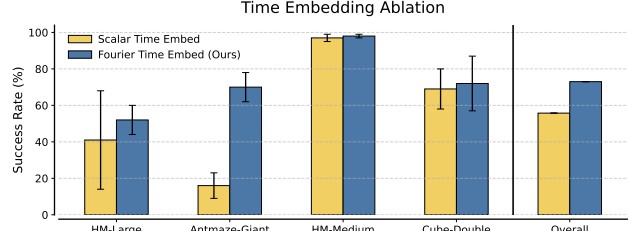

Figure 12: *Time embedding.* Replacing the Fourier-basis embedding of time with a scalar embedding results in significantly worse performance, highlighting the importance of Fourier features for conditioning on time.

ment). This means that choosing an interval $[l, u]$ with a non-trivial width is important. As discussed in Section 4.3, Figure 13 (right) shows that the curvature of the learned flow increases as the width of the interval grows. We measure curvature by computing the magnitude of the derivative of the velocity field as a function of time using finite differences. Concretely, we measured the expected value of $|dv_\theta(t, z(t))/dt|$ across state-action pairs in the offline dataset, averaged through training.

Putting results in Figure 13 together, we note that some degree of curvature is necessary for best performance, which is expected because otherwise, the flow collapses to behave like a monolithic critic. That said, excessive curvature makes the flow numerically harder to integrate, ultimately degrading performance. Based on these observations, we recommend practitioners use $\kappa := (u - l)/(Q_{\max} - Q_{\min})$ in the range of $\{0.1, 0.25\}$ as reliable starting points when tuning `floq` critic.

*2) Ablations for the time embedding.* In the default configuration of `floq`, we used a 64-dimensional Fourier embedding for the time $t$, provided as input to the velocity field (also see Dasari et. al Dasari et al. (2024) for a recent work training a diffusion policy also using Fourier embedding of $t$). As shown in Figure 12, replacing this Fourier embedding with a simple scalar embedding of $t$ leads to a significant drop in performance on several tasks. This highlights the importance of the Fourier embedding, which allow the velocity function to be meaningfully conditioned on $t$, enabling it to produce distinct behaviors at different integration times. Without such rich embeddings, the critic struggles to leverage temporal information effectively, and again collapse to the monolithic architecture. We therefore recommend that practitioners carefully utilize high-dimensional embeddings of time when using `floq`.

*3) How does the approach of embedding the interpolant $z(t)$ affect `floq` performance?* We observe that the approach of embedding $z(t)$ (Design Choice 2 in Section 4.3) plays a significant role in the performance of `floq`. As shown in Figure 14, HL-Gauss embeddings of $z(t)$ provide

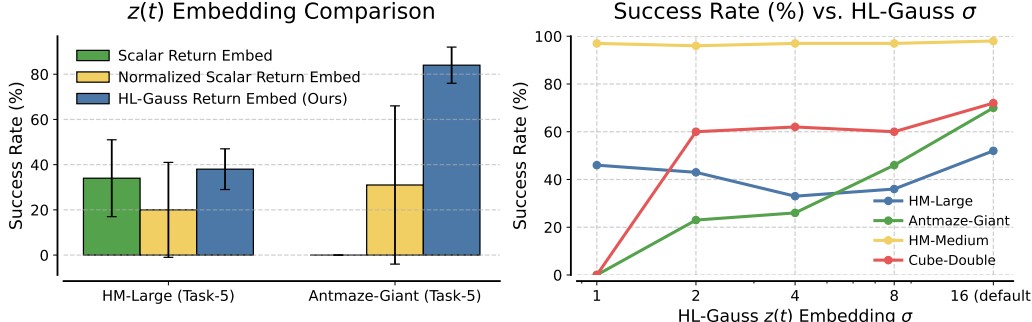

Figure 14: *Comparison of different approaches for representing the input interpolant in* `floq`. **Left**: performance on two representative tasks where that HL-Gauss embeddings outperform scalar and normalized scalar embeddings by reducing sensitivity to non-stationary inputs. **Right:** Ablation over HL-Gauss embedding scale $\sigma$ for the scalar flow interpolant input, showing that larger values provide broader bin coverage and stronger performance. Default $\sigma = 16.0$.

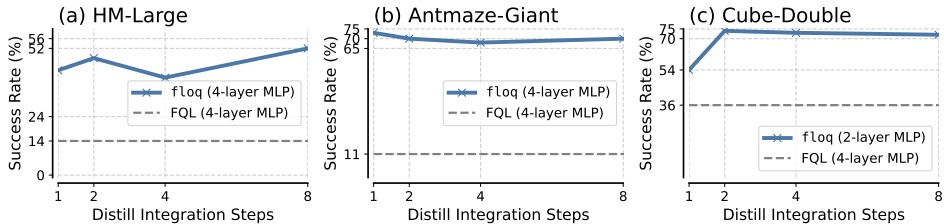

Figure 16: *Effect of the number of integration steps used for policy extraction on* `floq` *performance.* Even though computing the target values for TD-learning utilizes a fixed number of 8 integration steps, in this ablation we utilize a smaller number of steps for extracting the policy. Performance is more robust to the number of integration steps used for policy extraction, suggesting that as long as target integration is sufficiently accurate, few steps suffice for policy distillation.

a significant advantage over scalar or normalized scalar embeddings. In particular, across several tasks we found HL-Gauss embeddings (with a sufficiently large value of $\sigma$) to be essential for achieving strong performance, and Figure 14 (left) highlights two representative tasks in this category.

HL-Gauss embeddings with broader bin coverage helps reduce the sensitivity of the network to non-stationary inputs, thereby stabilizing training and improving performance. In domains such as HM-Large (Task 5), using raw scalar embeddings increases the training gradient norm by roughly $4\times$ compared to HL-Gauss ($2.5 \times 10^4$ to $10^5$), reflecting greater instability in optimization due to non-stationary conditioning (see Figure 15). While normalizing $z(t)$ helps on some tasks over using the raw value, we found that HL-Gauss embeddings generally gave the best performance. In our implementation of the velocity network, we use HL-Gauss embeddings with a default scale of $\sigma = 16.0$.

Figure 14 (right) shows an ablation over smaller values of $\sigma \in \{1.0, 2.0, 4.0, 8.0\}$. Observe that larger values of $\sigma$ consistently yield stronger performance. Intuitively, increasing $\sigma$ leads to broader bin coverage for the HL-Gauss distribu-

Training Grad Norm On HM-Large (Task 5)

Figure 15: *Training Grad Norm With and Without HL-Gauss Encoding.* In domains such as HM-Large (Task 5), using raw scalar embeddings increases the training gradient norm by roughly $4\times$ compared to HL-Gauss ($2.5 \times 10^4$ to $10^5$), reflecting greater instability in optimization due to non-stationary conditioning

tion (see Figure 2, right), which helps mitigate the non-stationarity of the range of $z(t)$ over the course of training with TD-learning. These results highlight that selecting sufficiently large embedding scales is important for stabilizing learning and achieving strong downstream performance.

*4) How does the number of critic flow steps used for the policy update affect the performance of* `floq`? We next investigate the effect of varying the number of integration steps used for calculating

the Q-value for the policy update. Since we build our algorithm on top of FQL, we implement the policy update by first distilling the values produced by the flow critic into a one-step, monolithic Q-function. Then the policy extraction procedure (akin to SAC+BC) maximize the values of this distilled critic subject to a behavioral cloning loss. Note that this approach essentially decouples the number of integration steps used to compute the TD-target and the number of integration steps for policy extraction. As shown in Figure 16, as long as the number of integration steps for computing the target value are fixed (to 8 in this case), the performance of `floq` is relatively robust to the number of integration steps used for the policy update. Contrast this with the sensitivity to the number of integration steps used for computing the TD-target observed in Figure 4. The results indicate that once the target integration steps are sufficiently large (here, 8), the policy can be effectively distilled even with a small number of integration steps.

## A.6 HYPERPARAMETERS AND ADDITIONAL DETAILS

In this section, we present some details for `floq` that we could not cover in the main paper, along with a pseudocode and a complete list of hyperparameters used by our approach.

---

**Algorithm 1** Critic Flow Matching (`floq`) in conjunction with FQL (Park et al., 2025d)

**Given:** offline dataset of transitions $\mathcal{D}$,
**Models:** a flow critic, $Q_\theta^{\text{FLOW}}(s, a, z)$, a distilled critic, $Q_\psi^{\text{distilled}}(s, a, z)$, a flow policy $\pi_\phi(\cdot|s)$, one-step policy $\mu_\omega(s, \cdot)$.

**function** $Q_\theta^{\text{FLOW}}(s, a, z)$ $\quad\quad\quad\quad\quad\quad\quad\quad\quad\quad\quad\quad$ ▷ Flow Q-function, introduced by `floq`
$\quad$ **for** $t = 0, 1, \ldots, K - 1$ **do**
$\quad\quad$ $z(t + 1) \leftarrow z(t) + {}^1\!/_K \cdot v_\theta\left({}^t\!/_K, z(t) \mid s, a\right)$ $\quad\quad\quad\quad$ ▷ Euler method, time $t$ is normalized
$\quad$ **return** $z(K)$

**function** $\pi_\phi(a|s)$ $\quad\quad\quad\quad$ ▷ Flow policy from FQL, though policy training is orthogonal to `floq`
$\quad$ **for** $t = 0, 1, \ldots, M - 1$ **do**
$\quad\quad$ Sample $x(0) \sim \mathcal{N}(0, I_d)$
$\quad\quad$ $x(t + 1) \leftarrow x(t) + {}^1\!/_M \cdot w_\phi\left({}^t\!/_M, x(t) \mid s\right)$ $\quad\quad\quad\quad$ ▷ Euler method, time t is normalized
$\quad$ **return** $x(M)$

**while** not converged **do**
$\quad$ Sample batch $\{(s, a, r, s')\} \sim \mathcal{D}$

$\quad$ ▷ Train vector field $v_\theta$ in flow critic $Q_\theta^{\text{FLOW}}$
$\quad$ $a' \leftarrow \text{Sample}(\pi_\phi(\cdot|s'))$ $\quad\quad\quad\quad$ ▷ Sample actions from policy, typically the one-step policy for FQL
$\quad$ $z(0) \sim \text{Unif}\,[l, u], z'(0)_{1:m} \sim \text{Unif}\,[l, u]$ $\quad\quad\quad\quad$ ▷ Sample initial noise for computing the Q-value
$\quad$ $z(1) \leftarrow r + \gamma \cdot {}^1\!/_m \cdot \sum_{i=1}^m Q_{\bar{\theta}}^{\text{FLOW}}(s', a', z_i'(0))$ $\quad\quad\quad$ ▷ Use noise $z_i'(0)$ for computing TD-target
$\quad$ $z(t) \leftarrow (1 - t) \cdot z(0) + t \cdot z(1)$ $\quad\quad\quad\quad$ ▷ Compute interpolant $z(t)$ for random $t$
$\quad$ Update $\theta$ to minimize $\mathbb{E}\left[(v_\theta\left(t, z(t) \mid s, a\right) - (z(1) - z(0)))^2\right]$ $\quad\quad$ ▷ Linear flow-matching loss

$\quad$ ▷ Train distill critic $Q_\psi^{\text{distill}}$ for policy extraction
$\quad$ Update $\psi$ to minimize $\mathbb{E}_{z(0)}\left[(Q_\psi^{\text{distill}}(s, a) - Q_\theta^{\text{FLOW}}(s, a, z(0)))^2\right]$

$\quad$ ▷ Train a BC flow policy $\pi_\phi$, analogous to FQL
$\quad$ $x(0) \sim \mathcal{N}(0, I_d)$
$\quad$ $x(1) \leftarrow a$
$\quad$ $t \sim \text{Unif}([0, 1])$
$\quad$ $x(t) \leftarrow (1 - t) \cdot x(0) + t \cdot x(1)$ $\quad\quad\quad\quad$ ▷ For FQL policy, compute policy interpolant
$\quad$ Update $\phi$ to minimize $\mathbb{E}\left[\|w_\phi\left(t, x(t)|s\right) - (x(1) - x(0))\|_2^2\right]$ $\quad\quad$ ▷ Flow-matching loss for policy

$\quad$ ▷ Train one-step policy $\mu_\omega$ to maximize the learned distill critic while staying close to BC flow policy
$\quad$ $x \sim \mathcal{N}(0, I_d)$
$\quad$ $a^\pi \leftarrow \mu_\omega(s, x)$
$\quad$ Update $\omega$ to minimize $\mathbb{E}\left[-Q_\psi^{\text{distill}}(s, a^\pi) + \alpha\|a^\pi - \pi_\phi(s, z)\|_2^2\right]$
**return** One-step policy $\pi_\omega$

---

**Hyperparameters for offline RL results.** Following Park et al. Park et al. (2025d), we tune the BC coefficient $\alpha$ on the `default-task` of each environment and then fix this value for the remaining tasks. For both FQL and `floq`, $\alpha$ is tuned over $\{\alpha_{\text{FQL}} - \Delta, \alpha_{\text{FQL}}, \alpha_{\text{FQL}} + \Delta\}$, where $\Delta = 100$

Table 6: **Hyperparameters for `floq`.** Differences from FQL are shown in light blue within brackets. Other hyperparameters are kept to be the same as FQL.

| Hyperparameter | Value (`floq`) |
|---|---|
| Learning rate | 0.0003 |
| Optimizer | Adam (Kingma & Ba, 2015 Kingma & Ba (2015)) |
| Gradient steps | 2M (Offline), 1M + 2M (Online FT) |
| Minibatch size | 256 (default), 512 for `hm-large`, `antmaze-giant` |
| Flow $Q$ Network MLP dims | [512,512,512] (default), [512,512] for `cube` envs |
| Distill $Q$ MLP dims | [512,512,512,512] (not used in FQL) |
| Nonlinearity | GELU (Hendrycks & Gimpel, 2016 Hendrycks & Gimpel (2016)) |
| Target network smoothing coeff. | 0.005 |
| Discount factor $\gamma$ | 0.99 (default), 0.995 for `antmaze-giant`, `humanoidmaze`, `antsoccer` |
| Flow time sampling distribution | Unif($[0, 1]$) |
| Clipped double Q-learning | False (default), True (`antmaze-giant`) (+ `antmaze-large` in FQL) |
| BC coefficient $\alpha$ | Tables 7, 8 |
| Actor Flow steps | 10 |
| Critic Flow steps | 8 (default), Table 9 for env-wise (not used in FQL) |
| Initial Sample Range | 0.1 (default), Table 9 for env-wise (not used in FQL) |
| Number Of Initial Noise Samples | 8 (not used in FQL) |
| Fourier Time Embed Dimension | 64 (not used in FQL) |

Table 7: Environment-wise BC-Coefficient ($\alpha$) for FQL and `floq` (Offline RL).

| Environment (5 tasks each) | $\alpha$ (FQL), $\alpha$ (`floq`) |
|---|---|
| `antmaze-large` | 10, 10 |
| `antmaze-giant` | 10, 10 |
| `hmmaze-medium` | 30, 30 |
| `hmmaze-large` | 30, 20 |
| `antsoccer-arena` | 10, 10 |
| `cube-single` | 300, 300 |
| `cube-double` | 300, 300 |
| `scene-play` | 300, 300 |
| `puzzle-3x3` | 1000, 1000 |
| `puzzle-4x4` | 1000, 1000 |

for the `puzzle`, `cube`, and `scene` environments, and $\Delta = 10$ for the `ant` and `humanoid` environments. The baseline values $\alpha_{\text{FQL}}$ are taken from Table 6 in Park et al. Park et al. (2025d), and the final values for both methods are reported in Table 7. For `floq`, after tuning $\alpha$ with the default configuration ($K = 8$ flow steps and width $(u - l) = \kappa(Q_{\max} - Q_{\min})$ with $\kappa = 0.1$), we tune $K \in \{4, 8\}$ and $\kappa \in \{0.1, 0.25\}$ on the `default-task` of each environment. These values, referred to as `floq`(Best), are reported in Table 9. In all cases, for `floq`, we utilize $m = 8$ samples of initial noise to compute the target Q-value as discussed in Section 4.2.

**Hyperparameters for online fine-tuning.** Most hyperparameters (unless otherwise stated) remain similar in online fine-tuning and offline RL pre-training. For both FQL and `floq`, $\alpha$ is tuned in the range $[10, 100]$ (step size 10) for the `ant` and `humanoid` environments, and in $[100, 1000]$ (step size 100) for the `cube`, `scene`, and `puzzle` environments. The selected $\alpha$ values are given in Table 8.

For `floq`, after tuning $\alpha$ with the default configuration ($K = 8$, $\kappa = 0.1$), we tune $K \in \{4, 8, 16\}$ and $\kappa \in \{0.1, 0.25\}$ per environment. The chosen values are reported in Table 10.

**Number of seeds.** We ran 3 seeds for each configuration of both `floq` and **FQL** on each task, for both offline RL and online fine-tuning.

In summary, we tuned the common hyperparameters for `floq`(Def.) and **FQL** the same amount on the default task for each environment (following Park et al. (2025c)). For `floq`(Best), we additionally tuned the `floq` specific hyper-parameters $K$ and $\kappa$ on the default task.

Table 8: Environment-wise BC-Coefficient ($\alpha$) for FQL and `floq` (Online Fine-Tuning).

| Environment (5 tasks each) | $\alpha$ (FQL), $\alpha$ (floq) |
|---|---|
| antmaze-large | $10, 10$ |
| antmaze-giant | $10, 10$ |
| hmmaze-medium | $80, 30$ |
| hmmaze-large | $40, 20$ |
| antsoccer-arena | $30, 30$ |
| cube-single | $300, 300$ |
| cube-double | $300, 300$ |
| scene-play | $300, 300$ |
| puzzle-3x3 | $1000, 1000$ |
| puzzle-4x4 | $1000, 1000$ |

Table 9: Environment-wise Initial Sample Range ($\frac{u-l}{Q_{\max}-Q_{\min}}$) and Flow Steps ($K$) for `floq` (Best) (Offline RL).

| Environment (5 tasks each) | ($\frac{u-l}{Q_{\max}-Q_{\min}}, K$) |
|---|---|
| antmaze-large | $(0.1, 8)$ |
| antmaze-giant | $(0.1, 4)$ |
| hmmaze-medium | $(0.1, 8)$ |
| hmmaze-large | $(0.1, 8)$ |
| antsoccer-arena | $(0.1, 8)$ |
| cube-single | $(0.1, 8)$ |
| cube-double | $(0.1, 8)$ |
| scene-play | $(0.1, 8)$ |
| puzzle-3x3 | $(0.1, 8)$ |
| puzzle-4x4 | $(0.25, 8)$ |

Table 10: Environment-wise Initial Sample Range ($\frac{u-l}{Q_{\max}-Q_{\min}}$) and Flow Steps ($K$) for `floq` (Online FT).

| Environment (5 tasks each) | ($\frac{u-l}{Q_{\max}-Q_{\min}}, K$) |
|---|---|
| antmaze-large | $(0.1, 8)$ |
| antmaze-giant | $(0.1, 4)$ |
| hmmaze-medium | $(0.1, 8)$ |
| hmmaze-large | $(0.1, 16)$ |
| antsoccer-arena | $(0.1, 8)$ |
| cube-single | $(0.1, 8)$ |
| cube-double | $(0.1, 8)$ |
| scene-play | $(0.1, 8)$ |
| puzzle-3x3 | $(0.1, 8)$ |
| puzzle-4x4 | $(0.25, 8)$ |

### A.7 FLOW VISUALIZATIONS

We visualize the evolution of the learned flow critic during training on `cube-double` in Figure 17, with $\kappa = 0.1$. Because raw Q-values can have large magnitudes, directly plotting them makes it difficult to assess the curvature of the learned flow. Instead, we plot advantage values, defined as the gap between the predicted Q-value obtained by integrating for $k$ flow steps at various noise samples $z_i(0)$, namely $\psi(k, z_i(k) \mid s, a)$ for $i \in [5], , k \in [1, \dots, K]$, and the expected value of that state–action pair after $K$ steps, scaled linearly to $k$ steps. Put simply, this advantage quantifies how far the intermediate estimate $\psi(k, z_i(k) \mid s, a)$ deviates from the "straight line" path between the initial noise sample $z_i(0)$ and the final Q-value. We find that these deviations are consistently non-zero and vary substantially across the integration process. In many cases, they exhibit a characteristic pattern of overshooting followed by correction: larger deviations early on that diminish as integration

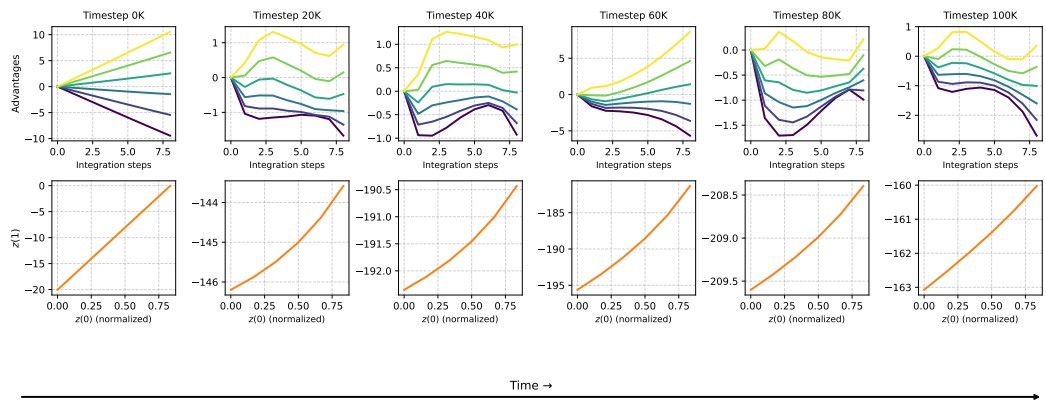

Figure 17: *Visualizing the evolution of the trajectories of the flow critic during training.*

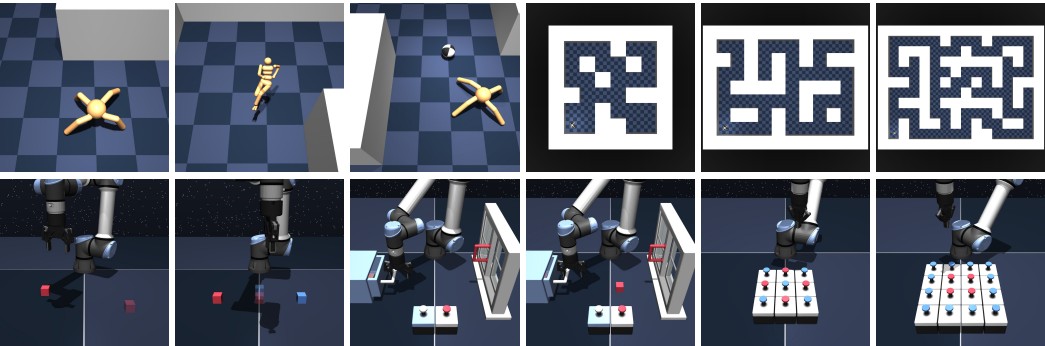

Figure 18: **OGBench (Park et al., 2025a) domains**. These tasks include high-dimensional state and action spaces, sparse rewards, stochasticity, as well as hierarchical structure.

proceeds. These dynamics provide direct evidence that the learned flows follow curved rather than linear trajectories. We also visualize the final Q-value output $z(1)$ as a function of the input $z(0)$ in Figure 17 (bottom) and find that the final $z(1)$ depends non-linearly on the initial noise value.

## A.8    ENVIRONMENT VISUALIZATIONS

We visualize OGBench tasks in Figure 18.

## A.9    WALL CLOCK RUN-TIME

We report the wall clock run-times for FQL and `floq` in Table 11.

Table 11: Total wall-clock runtime (in $10^3$ seconds) for FQL and `floq` with varying numbers of flow integration steps across four representative environments. Reported numbers correspond to 2M training steps.

| Environment | FQL | `floq` (Flow Steps) | | | | |
|---|---|---|---|---|---|---|
| | | 1 | 2 | 4 | 8 | 16 |
| HM-Maze Large | 14 | 24 | 28 | 35 | 50 | 79 |
| HM-Maze Medium | 12 | 19 | 21 | 23 | 30 | 47 |
| Cube-Double | 10 | 15 | 16 | 17 | 19 | 26 |
| Antmaze-Giant | 10 | 20 | 24 | 30 | 45 | 74 |

A.10    VARIANCE OF FLOW Q VALUE SAMPLES

To assess how much the predicted Q-values vary for a fixed $(s, a)$ when sampling different initial noise vectors $z$, we explicitly tracked the variance of the flow-matched Q-value samples during training. As shown in Figure 19, the variance decreases steadily over the course of training and eventually converges to a small value, consistent with the theoretical intent of representing a Dirac delta distribution.

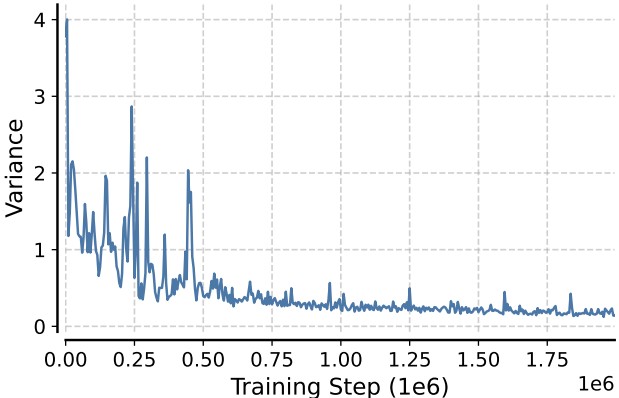

Figure 19: **Variance of Flow Q Value Samples.** The variance decreases steadily over the course of training, consistent with the theoretical intent of representing a Dirac distribution.

A.11    UNDERSTANDING THE PERFORMANCE DEGRADATION AT LARGE $K$ ON
          ANTMAZE-GIANT (FIGURE 6)

Observe from Figure 6 that we observe a degradation at large $K$ not just for `floq` but also for the monolithic ResNet approach. This makes us believe that the degradation in critic performance is largely not specific to `floq` but also applicable to a ResNet. Diagnostic experiments suggest that a degradation with larger $K$ stems from overfitting to very high capacity targets for both the ResNet and `floq` architecture.

For example, the best performing configuration on Antmaze-Giant ($K = 4$, see Figure 6) exhibits a higher TD error than the $K = 8$ or $K = 16$ settings (see Figure 20), whose lower TD errors coincides with worse control performance. This aligns with the view that excessively small TD error often indicates overfitting than better value estimation. In support of this, when we add a small amount of noise (uniformly sampled in $[-0.5, 0.5]$) to the TD target for the $K = 16$ critic, success rate increases by $7 - 10$ percent and the TD-error also increases to a healthier regime (see Figure 21). Taken together, these results point to a shared underlying cause: very high $K$ increases the critic capacity in a way that encourages overfitting to sharp targets, and moderate slack or noise improves the robustness needed for strong downstream performance.

A.12    EFFECT OF SIZE OF DISTILLED CRITIC

A natural question is whether the capacity of the distilled critic must scale with the number of flow integration steps $K$. Since larger $K$ produces a more expressive flow critic, an under-parameterized distilled critic may be unable to represent it faithfully, potentially degrading policy performance.

To study this, we performed a controlled experiment on the HMMAZE-LARGE task, evaluating a full cross-product of: (i) distilled critic depth: a 2-layer vs. 4-layer MLP (default), and (ii) number of flow steps: $K \in \{4, 8\}$. For all runs, the flow critic is trained via flow matching as usual, and the distilled critic is trained to regress onto the flow critic's Q-value (computed with $K$ integration steps).

Table 12 summarizes the results. We observe that increasing the depth of the distilled critic from two to four layers consistently improves performance. Moreover,the gap between the 2-layer and 4-layer variants widens substantially for $K = 8$. This indicates that using a small distilled critic effectively "bottlenecks" the expressivity of the flow critic when $K$ is large, confirming that distillation capacity must be sufficient to preserve the benefits of larger-flow-step training.

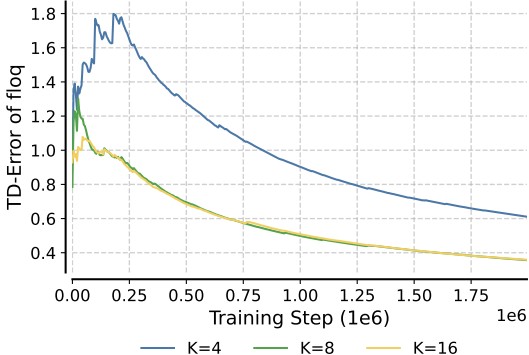

Figure 20: **TD-Error v/s Flow-Steps.** The best performing configuration on Antmaze-Giant ($K = 4$, see Figure 6) exhibits a higher TD error than the $K = 8$ or $K = 16$ settings, whose lower TD errors coincides with worse control performance (see Figure 6). This aligns with the view that excessively small TD error often indicates overfitting than better value estimation.

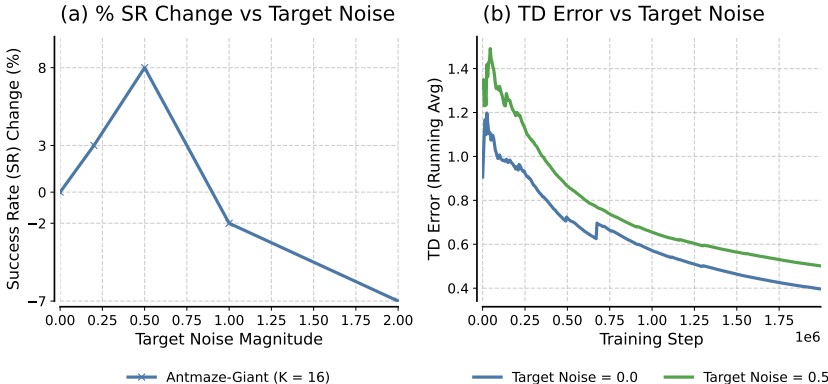

Figure 21: **Effect of Target Noise on Success Rate And TD-Error (Antmaze-Giant,** $K = 16$**).** Effect of target noise magnitude on (a) the change in success rate and (b) the TD error during training. We see that small amounts of target noise (magnitude $\approx 0.5$) improves performance by $7 - 10$ percent and also increases TD-error to a healthier regime. However, too much target noise degrades performance, as expected.

Table 12: **Effect of distilled-critic capacity for different flow-step on HMMAZE-LARGE.**
A shallow (2-layer) distilled critic performs worse than a 4-layer critic, especially for larger $K$, suggesting that insufficient capacity limits the distilled critic's ability to represent the more expressive Q-function produced by higher flow-step integration.

| Distilled Critic | 4 Flow Steps | 8 Flow Steps |
|---|---|---|
| 2-layer MLP | $20 \pm 4$ | $26 \pm 3$ |
| 4-layer MLP | $24 \pm 4$ | $52 \pm 8$ |

## A.13 DISCUSSION, CONCLUSION, AND FUTURE WORK

This work introduced `floq`, a flow-matching approach to training critics that scales Q-function capacity through iterative integration and dense supervision, achieving state-of-the-art offline RL results and online fine-tuning results. Future directions include understanding how to set integration steps, exploiting `floq`'s cascaded family of critics for efficiency, and combining sequential test-time scaling with ensembles. Theoretically, it is important to study how curved flows enable error correction when learning Q-values for TD-learning.

