# OpenReview forum: "floq: Training Critics via Flow-Matching for Scaling Compute in Value-Based RL"
_ICLR.cc/2026/Conference — ICLR 2026 Poster_

### Official Review · Reviewer_Jc59 · 2025-10-20

**Soundness:** 2
**Presentation:** 2
**Contribution:** 3
**Rating:** 6
**Confidence:** 3

**Summary:**

The paper proposes a novel way of applying flow mathching in offline RL by predicting the Q-value function with TD learning: for that they introduce a latent variable $z$ that at $t=0$ is sampled from a uniform distribution and by following the learned velocity $v_{\theta}(t, z)$ we should obtain $\delta(Q(s,a))$  at $t = 1$. To learn the velocity field $v_{\theta}(t,z)$, this approach suggests using the interpolant between the initial noise $z$ and TD-bootstrapped Q function at every step of numerical integration of the flow.  The paper builds on top of the previous work that considered flow matching for policy learning and showed that using flow matching for Q learning can bring up to x1.8 better results on some hard tasks from OGBench, where hardness is measured with respect to FQL baseline. This approach is contrasted to monolithic way of learning Q and over the series of ablation studies, authors demonstrate that this improvement cannot be attributed to just an increase of compute. Empirically, the authors show that the main ingredients to make flow matching work for TD learning are 1) the careful choice of support interval for $z$ , 2) using categorical representation for the latent input $z(t)$ with HL-Gauss encoding of high variance.

The overall idea is original and results in a clear improvement over the monolithic way of learning Q-function, but more explanation/investigation is needed to understand the degraded performance when large $K$ are considered. Moreover, the paper can improve its writing, while providing more details on the experimental setup in the main paper. For all these reasons, I recommend a weak accept.

**Strengths:**

The idea of using flow matching for TD learning is very interesting and further extends the works on adapting flow matching techniques to RL. The paper shows that this approach first of all allows us to reach new state-of-the-art results for many tasks from OBbench constrained to a single task. Secondly, they showed that this algorithm proposes a good way to further scale offline RL and it is compared to other ways of scaling such as increasing the compute power in sequential way (comparison to ResNet) and in parallel way (comparison to ensemble methods). The authors also perform a vast ablation study to justify their design choices and discovering a set of default hyper parameters that work uniformly well for all tasks considered in the experiments.

**Weaknesses:**

I found that a lot of experimental details were not clear enough or transparent while reading the paper. What is success-rate in OGbench? Is success-rate the only metric, because in Figure 7 you mention a generic performance metric. How do you extract policy from the learned Q-furnction? Those details become more clear after consulting the appendix, but a short summary in the main paper is necessary. Moreover, it is only upon reading the appendix, it becomes clear that floq essentially builds on top of FQL and shares most of its implementation details except for training the Q function, which in FQL is done in monolithic way. Thus, in practice floq needs to distill its learned Q function from flow matching into a single MLP that can be used to obtain approximated Q in a single step. In this regard, I would like the authors to clarify whether they use a distilled Q function for all their experimental results or they use sometimes a Q function obtained directly from flow matching.

Moreover, I found the study performed in Section 5.3 in the point 1) a bit unsatisfactory. First of all, I would like some more details on this experiment’s setup: does $K$ correspond to a number of integration steps used during the training and then the results are reported for the distilled Q function? From Figure 4, we see that using larger number of $K$ leads to a significant degradation on one of the tasks, and the only explanation that authors provide is that it might be overfitting, but no further analysis to confirm this hypothesis is performed. In this case, why cannot be underfitting? To my mind, you might need different number of training steps to train floq with small $K$ vs large $K$. Or you might need to consider larger networks for distillation to represent floq with $K>8$. I believe this question should be further explored, as, on the first glance, the choice of $K$ should only affect the granularity of the flow and thus having worse performance with larger $K$ is contra-intuitive.

Another possible limitation is that the authors only considered OGbench for their experiments, which is a dataset designed originally for goal-conditioned offline RL. The authors argue that the most common dataset such as D4rl has been saturated, but in this regard, why not considering d5rl for example?

Finally, the authors only test their algorithm on 3 seeds, which is not enough to get reliable statistics, especially for multi-agent comparison https://arxiv.org/abs/2306.10882 (see App.G). Even if it is just for a few tasks, but performing comparisons with more seeds is important for more conclusive results.

Minor:
- L138 remove the in front of our
- L158 refer to the papers that use averaging/doing ensembles for performing TD
- Section 5.3: the order of the questions is non-linear: point 2) is after 3)
- L1025: helps TO mitigate



Weird sentences:
- L194-195: Interestingly, we find that this problem can is a result of two peculiarities associated with applying flow-matching to TD
- L235: When we then run (imperfect) TD-loss training on these interpolants,
- L227-229: When the width of the interval $[l,u]$ is small, and the overlap between this interval and the range of target $Q$-values we hope to see is minimal, we would expect to see more straight flow traversals, that might be independent of interpolant $z$.
- L262-263: We show in our experiments THAN doing so helps substantially by encouraging the network to meaningfully utilize this time.
- L437: To answer this question, we trained a variant of floq where we supervised the velocity field was supervised only at t= 0 and used only a single flow step.
- L444: oneintegration

**Questions:**

See weaknesses. More questions:

  1. Can floq be adjusted to distributional RL setting, as currently flow matching is used to approximate a dirak distribution over $Q(s,a)$, while it can be seen as a natural framework to learn distributions over $Q$?
2.  How do you measure Q_min and Q_max?
3. What is a difference between default tasks and all tasks in OGbench?
4. (L430-431) Can floq be parallelised though? Because, even if the cumulative compute time is the same, when it is parallizable you can expect that you can gain some speedup, so the actual time will reduce roughly by number of processes P.
5. How online fine-tuning is performed, how the data is collected for online finetuning?
6. If flow matching is done with an ODE integrator why not to try different integration schemes? According to you, can it remedy the degradation of the performance observed for the cases with the high curvature.
7. How different are $Q(s,a)$ for fixed $s$ and $a$, but with different initial $z$? Is there a high discrepancy, even if they are trained to represent dirak distribution?

---

> ### Author Response · Authors · 2025-11-23
>
> Thank you for your positive review and the feedback! We address your concerns below by presenting new experiments to understand the role of distillation for critic training with floq and the degradation in performance with both floq and ResNets as we scale the number of integration steps / layers in the ResNet. We provide several clarifications regarding the experimental setup both in the response below and the updated version of the paper as well.
>
> **Please let us know if our responses address your concerns, and if so, we would be grateful if you are willing to raise your score.**
>
> ___
>
> ## Clarification Questions
>
> ### >  **How do you extract policy from the learned Q-function? Those details become more clear after consulting the appendix, but a short summary in the main paper is necessary.**
>
> You are right that we use most of the machinery from FQL for policy extraction, though we have added new experiments with ReBRAC, that use a Gaussian policy and do not need the FQL machinery (distilled one-step policy) and we find that floq still outperforms the base ReBRAC algorithm as shown  in **Table 4, Appendix A.4**, indicating that the machinery behind FQL’s policy extraction is not that crucial for performance.
>
> That said, we have now added a description of the policy learning process to the main body to address this concern (see **Section 4.3, last paragraph**).
>
> ---
>
> ### > **Thus, in practice floq needs to distill its learned Q function from flow matching into a single MLP that can be used to obtain approximated Q in a single step. I would like the authors to clarify whether they use a distilled Q function for all their experimental results or they use sometimes a Q function obtained directly from flow matching.**
>
> This is a great question—thank you for raising it. For all of our experiments in the submission, we distill the flow-based Q-function into a monolithic critic. This allows us to compute the action-gradient of $Q_\theta(s,a)$ efficiently and stably, in the same way that FQL also distills its flow-based policy into a one-step policy network for actor updates (see **Section 4.3, last paragraph** and the **Algorithm box 1 in Appendix A.7**).
>
> We also tried taking the actor action gradients directly through the flow integrated Q-values, but this approach performed substantially worse because of recursive backpropogation through time. For this reason, we rely on a distilled critic for all reported results.
>
> ---
>
> ### > **Does $K$ correspond to a number of integration steps used during the training and then the results are reported for the distilled Q function?**
>
> Yes, K refers to the number of integration steps for the flow Q-function during training. We use K integration steps for also computing the target values, and for computing the Q-value that we use to distill the target Q-function into the distilled network.
>
> ---

---

> ### Author Response · Authors · 2025-11-23
>
> ## Experimental Questions
>
> ### > **From Figure 4, we see that using larger number of flow steps  leads to a significant degradation on one of the tasks, and the only explanation that authors provide is that it might be overfitting, but no further analysis to confirm this hypothesis is performed. In this case, why cannot be underfitting? To my mind, you might need different number of training steps to train floq with small  vs large.**
>
> This is a great question! Please note that we observe a degradation at a higher $K$ not just for floq but also for the monolithic ResNet approach. This makes us believe that a degradation in the critic performance is largely not specific to floq but is also applicable for a ResNet. We are still running several diagnostic experiments to evaluate floq, but we have some results that indicate that  a degradation with larger $K$ stems from **overfitting to very high capacity targets** with both the ResNet and the floq architecture.
>
> For example, the best-performing configuration $(K=4)$ exhibits a higher TD error than the $K=8$ or $K=16$ settings, whose lower TD errors coincide with worse control performance. This aligns with the view that excessively small TD error often indicates overfitting rather than better value estimation. Supporting this, when we add a small amount of Gaussian noise to the TD target for the $K=16$ critic, performance improves by about $7$ to $10$ percent and the TD error increases to match the healthier regime of $K=4$. Taken together, these results point to a shared underlying cause: very high $K$ increases the critic capacity in a way that encourages overfitting to sharp targets, and moderate slack or noise helps maintain the robustness needed for strong downstream performance.
>
> ---
>
> ### >  **Another possible limitation is that the authors only considered OGbench for their experiments, which is a dataset designed originally for goal-conditioned offline RL. The authors argue that the most common dataset such as D4rl has been saturated, but in this regard, why not considering d5rl for example?**
>
> In response to the other reviewers, we have now added results comparing  floq + ReBRAC to ReBRAC (where we observe better performance with floq, see **Appendix A.4**), and floq to FQL on the easier D4RL tasks, where they perform comparably (see **Appendix A.5**).
>
> We initially chose to use OGBench because our main comparison FQL (Park et al., 2025) creates single-task versions of OG Bench environments and uses them for standardized comparisons. Since then, many papers have used these single-task versions of OGBench tasks for benchmarking, including several of our comparisons (FQL, SORL, DSRL).
>
> We chose to not compare on D5RL so far, though we will attempt to do this for the final version, primarily because we have not seen adoption of that benchmark in the community. None of our baseline comparisons or base algorithms evaluate on D5RL and the benchmarking tables in the D5RL paper are not fully complete. That said, we plan to set up the visual cube, scene and puzzle tasks from FQL as an additional point of comparison, and would appreciate any more feedback you might have on comparison benchmarks or tasks.
>
> ---
>
> ### > **Finally, the authors only test their algorithm on 3 seeds, which is not enough to get reliable statistics, especially for multi-agent comparison https://arxiv.org/abs/2306.10882 (see App.G). Even if it is just for a few tasks, but performing comparisons with more seeds is important for more conclusive results.**
>
> We are running additional seeds and will include these expanded results in the updated version of the paper. We also note that our evaluation already benefits from the many-task regime: OGBench provides 5 single-task variants per environment (50 tasks in total), giving us 150 total runs per algorithm (3 seeds × 50 tasks).
>
> That said, as discussed in Agarwal et al. (2023) and the accompanying rliable framework, IQM and performance profile evaluations with few seeds remain statistically meaningful when taken across many tasks, even though each task presents a fewer number of seeds. **The rliable metrics that we already reported in the Appendix A.3 (Figure 8 and 9)**, show that floq consistently and significantly outperforms FQL across the 50 OGBench tasks.
>
> In addition, the paper cited by the reviewer focuses on reliability issues that arise when comparing three or more algorithms. In our case, we primarily compare floq against a single main baseline (FQL), and rliable provides strong statistical confidence for such pairwise comparisons. We infact also plot the P(floq > FQL) using rliable, and this shows a clear trend in favor of floq.

---

> ### Author Response · Authors · 2025-11-23
>
> ### > **Can floq be adjusted to distributional RL setting, as currently flow matching is used to approximate a dirak distribution over , while it can be seen as a natural framework to learn distributions over ?**
>
> Combining flow-based Q-functions with a distributional RL update (e.g., quantile regression or categorical projection) is an interesting direction, and we plan to explore it in future work.
>
> We would also like to emphasize that even though floq uses a simple expected value TD backup (thus is not a distributional RL algorithm), it still outperforms representative distributional RL baselines such as IQN and C51 (see **Table 2 in Appendix A.3** in the paper).
>
> ---
>
> ## Minor Clarifications
>
> ### > **How do you measure Q_min and Q_max?**
>
> We obtain Q_min and Q_max by computing the minimum and maximum empirical returns in the offline dataset. During online learning, we calculate online returns and update Q_min and Q_max, similarly to previous works using categorical critics [1,2]
>
> [1] Bigger, Regularized, Categorical (BRC), High Capacity Value Functions Are Efficient Multi-Task Learners, Nauman et. al, 2025. [2] Hyperspherical Normalization for Scalable Deep RL (SimbaV2), Lee et. al, 2025.
>
> ### > **What is success-rate in OGbench? Is success-rate the only metric, because in Figure 7 you mention a generic performance metric.**
>
> Thanks for pointing out this typo. The graph indeed shows success-rates, but we labeled the y-axis wrongly. We fixed this error in the new version of our manuscript (see **Figures 7,12,13,14**).
>
> ### > **What is a difference between default tasks and all tasks in OGbench?**
>
> In OGBench each environment provides five single-task variants, from which the benchmark designates one as the default task. The **default task** is used as the canonical representative of the environment, while **"all tasks"** refers to evaluating on the full set of five single-task variants. In our paper, we report the main results on all tasks (see **Table 1**). We use the default task only for auxiliary comparisons, ablations, and diagnostic experiments where using a single representative task is standard practice.
>
> Following the convention used in the FQL paper, we also perform all hyperparameter tuning on the default tasks, and we then apply the same hyperparameters across all tasks within the same environment. This matches the intended evaluation protocol in OGBench.
>
> ### >  **(L430-431) Can floq be parallelised though? Because, even if the cumulative compute time is the same, when it is parallizable you can expect that you can gain some speedup, so the actual time will reduce roughly by number of processes P?**
>
> floq is parallelizable in the same way as standard monolithic critics. In fact, throughout the paper, we already trained an **ensemble of 2 floq critics**. Increasing the ensemble size of floq critics should yield performance improvements analogous to increasing ensembles of monolithic critics.
>
> Importantly, our ablation comparing floq to ensembles of monolithic critics demonstrates that even when giving monolithic critics increased parallel compute (e.g., 4× or 8× ensembles), floq still substantially outperforms them. Thus, the gains from parallel compute of monolithic critics do not close the performance gap observed in our experiments.
>
> ### > **How online fine-tuning is performed, how the data is collected for online finetuning?**
>
> Our offline → online fine-tuning procedure follows the standard setup used in the FQL paper. We initialize the online replay buffer with the entire offline dataset, and during online interaction we append newly collected transitions to this buffer. The agent is updated continuously from this growing buffer without re-weighting the offline portion. As in FQL, we use a UTD ratio of 1, meaning that for each environment step we perform a single gradient update. This ensures a fair comparison to prior work.
>
> ### > **How different are Q  for fixed s  and a, but with different initial z? Is there a high discrepancy, even if they are trained to represent dirak distribution?**
>
> To assess how much the predicted Q-values vary for a fixed (s,a) when sampling different initial noise vectors $\mathbf{z}$, we explicitly tracked the variance of the flow-matched Q-value samples during training. As discussed in **Appendix A.11 (see Fig. 19)**, the variance decreases steadily over the course of training and eventually converges to a small value, consistent with the theoretical intent of representing a dirac distribution.
>
> ---
>
> We are happy to answer any further questions. Thank you again for helping us to improve our manuscript. We hope that the clarifications paired with additional experiments increase your confidence in our work. If so, we kindly ask to update the score accordingly.

---

> ### Comment · Reviewer_Jc59 · 2025-11-26
> **Follow-up. Further question on large K overfitting.**
>
> I thank authors for answering most of my questions and providing further clarifications. Indeed, your answers cover almost all my concerns and helped better to understand the merits of the proposed method.
>
> To return to the question of overfitting for large K, your arguments indeed indicate a possible overfitting to target Q values, have you included those tests (adding gaussian noise to targets for K=16 case) in the revised paper?
>
> Despite that, I still suspect that if you don’t adapt the size of distilled network to the parameter K, then the degradation of policy's performance can be expected, i.e. you lose your expressivity obtained with large K and thus degrade the performance. In general, have you tried to experiment with the size of the distilled neural network?

---

> > ### Author Response · Authors · 2025-11-28
> >
> > Thank you for the thoughtful follow-up questions and for the positive feedback on our earlier clarifications. We address each point below.
> >
> > ---
> >
> > ### **1. Added analysis for the K=16 degradation**
> > As suggested, we have now added the full set of experiments analyzing the degradation for $K=16$. Specifically, **Appendix A.12** of the revised paper now includes:
> >
> > - **TD-error curves** showing that the $K=4$ model achieves higher TD-error than $K=8$ and $K=16$ (**Figure 20**).
> > - **Experiments where we inject small noise into the TD targets**, which improves the $K=16$ performance (**Figure 21**).
> > ---
> > ### **2. Effect of distilled critic capacity**
> > We appreciate the insight regarding the relationship between the number of flow steps $K$ and the capacity of the distilled critic. To evaluate this directly, we ran a new set of experiments on the **Hmmaze-Large** environment using a cross-product of:
> >
> > - **Distilled critic depth:** 2-layer vs. 4-layer MLP (**default**)
> > - **Flow critic integration steps:** $K=4$ vs. $K=8$
> >
> > The results (mean ± std over 3 seeds) are shown below:
> >
> > #### **Performance of Distilled Critic vs. K on Hmmaze-Large**
> >
> > | Distilled Critic | 4 Flow Steps | 8 Flow Steps |
> > |------------------|--------------|--------------|
> > | **2-layer MLP**  | 20 ± 4       | 26 ± 3       |
> > | **4-layer MLP**  | 24 ± 4       | 52 ± 8       |
> >
> > These experiments suggest that:
> >
> > 1. **Smaller distilled critics consistently underperform**, confirming insufficient expressivity becomes a bottleneck.
> > 2. **The performance gap widens at larger $K$**, supporting the hypothesis that a low-capacity distilled critic cannot adequately capture the higher-expressivity flow Q-function produced with more integration steps.
> >
> > We have incorporated these results and discussion into **Appendix A.13** and **Table 12** of the revised paper.
> >
> > ---
> >
> > We hope these additions resolve all remaining concerns. If so, we would be very grateful if the score could be updated accordingly. Thank you again for the constructive and detailed feedback.

---

### Official Review · Reviewer_eV3j · 2025-10-30

**Soundness:** 2
**Presentation:** 4
**Contribution:** 2
**Rating:** 4
**Confidence:** 3

**Summary:**

This paper proposes *floq*, a flow-matching based Q-value estimation approach for offline and offline-to-online reinforcement learning. More specifically, it computes Q-values by iteratively running numerical integration of flow-matching's velocity field, allowing more fine-grained control and scaling than traditional monolithic architectures. The proposed method is evaluated on multiply OGBench tasks.

**Strengths:**

- The overall paper is well written and organized, and some illustrations are clear.
- The proposed flow-matching-based Q-learning is interesting.
- The results are competitive, and the authors provide comprehensive analysis on their design choices.

**Weaknesses:**

- I can understand that applying flow-match or diffusion models can improve policy learning due to the complex, multi-modal/-peak distribution of policies (e.g., you can take different actions at a state but still get similar rewards). However, when it comes to the Q function, I am confused that if Q-value is also multimodal given a certain action.
- The authors state that previous diffusion-based methods require supervision at every iterative step. But this can be alleviated by directly approximating the final Q-value from noise prediction, similar to DDIM.
- It would be better to also add the flow-matching to preliminary to help readers understand the following method smoothly.
- Although OGBench tasks are generally more challenging than standard D4RL, the comparison is still needed to illustrate the robustness of *floq*. In addition, more diffusion-based approaches should also be added to the comparison.
- There are repeated citations for the same "flow q-learning" paper.

**Questions:**

- What is the $m$ in the TD-target equation $y(s, a)$ in line 179.
- The authors state that Q_min and Q_max denote the minimal and maximal possible Q-value achievable on the task. But how do you calculate them in practice?

---

> ### Author Response · Authors · 2025-11-23
>
> Thank you for your review! To address your concerns, we have added new experiments comparing floq and baselines on D4RL, as well as experiments comparing floq with Gaussian policies in **Appendix A.4 and Appendix A.5** -- in both cases, we find that floq outperforms or matches these comparisons and attains the best overall results. We also clarify questions regarding “multimodality” of the Q-function, but believe that these might be stemming _potentially_ from some misunderstanding about the paper. We have also addressed other concerns by making updates to the paper. **Please let us know if these responses and updates to the paper address your concerns, and if so, it would be great if you are willing to accept the paper.**
>
> ---
>
> ### > **D4RL And The Robustness of floq**
>
> | **Environment**                   | **FQL**         | **floq**        |
> |------------------------------|-------------|-------------|
> | antmaze-large-diverse-v2     | 85 ± 5      | 82 ± 5      |
> | antmaze-large-play-v2        | 82 ± 8      | 75 ± 8      |
> | antmaze-medium-diverse-v2    | 73 ± 6      | 71 ± 10     |
> | antmaze-medium-play-v2       | 76 ± 7      | 80 ± 7      |
> | antmaze-umaze-diverse-v2     | 85 ± 7      | 78 ± 17     |
> | antmaze-umaze-v2             | 96 ± 2      | 97 ± 2      |
> | pen-human-v1                 | 51 ± 10     | 58 ± 9      |
> | pen-cloned-v1                | 78 ± 8      | 72 ± 5      |
> | pen-expert-v1                | 140 ± 5     | 140 ± 8     |
> | door-human-v1                | 0 ± 0       | 0 ± 0       |
> | door-cloned-v1               | 3 ± 1       | 3 ± 2       |
> | door-expert-v1               | 104 ± 0     | 104 ± 0     |
> | hammer-human-v1              | 1 ± 1       | 1 ± 1       |
> | hammer-cloned-v1             | 15 ± 13     | 10 ± 8      |
> | hammer-expert-v1             | 125 ± 3     | 125 ± 2     |
> | relocate-human-v1            | 0 ± 0       | 0 ± 0       |
> | relocate-cloned-v1           | 0 ± 0       | 0 ± 0       |
> | relocate-expert-v1           | 106 ± 2     | 108 ± 2     |
>
> As requested, we are now adding new results evaluating floq on the standard D4RL benchmark and comparing it directly against the FQL baseline. The results (now included in the paper as **Table 5 and Figure 11 in Appendix A.5** in the paper) show that floq performs similarly to FQL across all D4RL tasks, with **nearly identical** mean, median  and  IQM score (see **Figure 11 in Appendix A.5**). These findings confirm that floq does not sacrifice performance on simpler benchmarks.
>
> We believe that a similar performance of floq and FQL on D4RL is a sign of **saturation of the D4RL benchmark**, and a growing trend in the community to move away from these tasks [1,2, 3]: these tasks typically do not quite high capacity policies or critics, and have been optimized on substantially. Nonetheless, floq performs very well on these tasks.
>
> In addition, to further test whether the floq critic can improve other base RL algorithms, we paired the **floq critic with the ReBRAC Gaussian policy**, and the results (added in **Table 4 in Appendix A.4**) show that **floq substantially improves ReBRAC** (**average score: 30 → 41**). This demonstrates that the benefits of the floq critic extend beyond a specific actor architecture, and that it can be used as a drop-in improvement.
>
> |Environment       | **ReBRAC**        | **floq (ReBRAC)** | **FQL**           | **floq (FQL)**    |
> |-------------------|---------------|---------------|---------------|---------------|
> | Antmaze-Large     | 95 ± 4        | 97 ± 3        | 85 ± 4        | 94 ± 4        |
> | Antmaze-Giant     | 27 ± 22       | 91 ± 6        | 14 ± 20       | 70 ± 8        |
> | Hmmaze-Medium     | 16 ± 9        | 57 ± 12       | 58 ± 25       | 98 ± 1        |
> | Hmmaze-Large      | 2 ± 1         | 0 ± 1         | 14 ± 10       | 52 ± 8        |
> | Antsoccer-Arena   | 0 ± 0         | 1 ± 2         | 49 ± 11       | 49 ± 10       |
> | Cube-Single       | 92 ± 4        | 89 ± 4        | 94 ± 5        | 99 ± 2        |
> | Cube-Double       | 7 ± 3         | 3 ± 3         | 36 ± 6        | 72 ± 15       |
> | Scene-Play        | 50 ± 13       | 58 ± 8        | 78 ± 7        | 83 ± 10       |
> | Puzzle-3x3        | 2 ± 1         | 9 ± 6         | 16 ± 5        | 17 ± 6        |
> | Puzzle-4x4        | 10 ± 3        | 6 ± 4         | 11 ± 3        | 12 ± 4        |
> | **Average Score** | **30**        | **41**        | **45**        | **64**        |
>
> [1]  OGBench, Benchmarking Offline Goal Conditioned RL, Park et. al,  2025.
> [2] D5RL, Diverse Datasets for Data Driven Deep RL, Rafailov et. al, 2024.
> [3] The generalization gap in offline RL, Mediratta et. al, 2024.

---

> ### Author Response · Authors · 2025-11-23
>
> ### > **When it comes to the Q-function, I am confused if the Q-value is also multimodal given a certain action.**
>
> Thanks for raising this question. We believe there may be a misunderstanding here, and we would like to clarify. If this does not address your concern, please let us know.
>
> A key distinction is that, unlike policy distributions (which can indeed be multimodal), a Q-function is a scalar function of the input state and action. While the Q-function can have a complex and highly non-linear manifold over the space of states and actions, for any fixed state and action it evaluates to a single scalar. Therefore it is not multimodal in the usual probabilistic sense.
>
> Our use of flow matching is **not intended to model a multimodal target distribution**. Instead, it provides a convenient mechanism for **iterative computation of the Q-value** by integrating toward a Dirac-delta distribution at the target value. In principle, flow matching is not strictly required, but it offers a clean and scalable way to structure this iterative refinement. As shown in Figure 7 and discussion in **Section 5.3 (point 4)**, the performance gains arise from this iterative computation combined with the representational benefits of training the Q-function across different noise magnitudes (i.e., when using noise + single step of integration, we can still outperform a monolithic critic). This allows the model to spend more effective compute on estimating Q, which leads to consistent improvements.
>
> ---
>
> ### > **The authors state that previous diffusion-based methods require supervision at every iterative step. But this can be alleviated by directly approximating the final Q-value from noise prediction, similar to DDIM.**
>
> We believe there may be a misunderstanding. Our claim is not that prior diffusion-based approaches suffer because they require supervision at every integration step, nor that this supervision is inherently problematic. Instead, our point is that **providing supervision at intermediate stages of a deep computation graph enables better use of iterative computation** and improves performance. **Figure 6** and the discussion in **Section 5.3 (point 2)**  illustrates this: a ResNet that uses the same number of iterative steps performs worse than an equivalent floq model, and the only difference is that floq receives supervision at all intermediate steps.
>
> We suspect some of the confusion may arise from the assumption that we are modeling a probability distribution, similar to diffusion-based policy models. We are not modeling a policy distribution, and we are also not using a stochastic diffusion process. Our method uses a fully deterministic flow-matching style integration. Although one could in principle bypass intermediate integration steps and directly approximate the final Q-value (in a manner similar to DDIM), our claim concerns the benefits of supervising intermediate computations during training, which DDIM also uses.
>
> Please let us know if this clarifies the issue or if further explanation would be helpful.

---

> > ### Author Response · Authors · 2025-11-23
> >
> > ### > **In addition, more diffusion based approaches should also be added to the comparison.**
> >
> > Please note that to the best of our knowledge we are unaware of any prior work that utilizes flow-matching or diffusion-based modeling to represent the Q-function. Prior work only utilizes diffusion and flow-matching to model the policy distribution. Our experiments already include comparisons to the best of these approaches: flow-matching policies (e.g. FQL, which is already a state-of-the-art method using flow-matching policies and SORL) and diffusion-based (e.g. DSRL) approaches (see **Table 1 and Table 2**). We do also find that floq outperforms these approaches by  $\geq 1.8$ times.
> >
> > Since there is no prior work that applies flow-matching or diffusion style training to the Q-function, we compare to works that instead model the Q-value distribution and capture “multimodal” Q-value distributions, i.e., IQN and C51 in **Table 2, Appendix A.3**. Again we observe that floq outperforms IQN and C51 by $\geq 2$ times, which in turn performs at best similar to the FQL approach.
> >
> > | **Environment**        | **FQL**          | **FQL-IQN**      | **FQL-C51**      | **floq (Def.)**   | **floq (Best)**  |
> > |--------------------|--------------|--------------|--------------|----------------|--------------|
> > | Antmaze-Large      |  85 ± 4      |  86 ± 1      |  79 ± 3      | **94 ± 4**     | **94 ± 4**   |
> > | Antmaze-Giant      |  14 ± 29     |   5 ± 6      |   0 ± 0      |  70 ± 8        | **86 ± 4**   |
> > | Hmmaze-Medium      |  58 ± 25     |  27 ± 17     |  23 ± 16     | **98 ± 1**     | **98 ± 1**   |
> > | Hmmaze-Large       |  14 ± 10     |  22 ± 5      |   6 ± 2      | **52 ± 8**     | **52 ± 8**   |
> > | Antsoccer-Arena    | **49 ± 11**  |  44 ± 3      |  36 ± 7      |  49 ± 10       |  49 ± 10     |
> > | Cube-Single        |  94 ± 5      | **98 ± 1**   | **98 ± 1**   |  99 ± 2        | **99 ± 2**   |
> > | Cube-Double        |  29 ± 8      |  57 ± 2      |  40 ± 11     | **72 ± 15**    | **72 ± 15**  |
> > | Scene-Play         |  78 ± 7      |  80 ± 4      | **82 ± 11**  |  83 ± 10       | **83 ± 10**  |
> > | Puzzle-3×3         |  14 ± 4      | **20 ± 3**   |  13 ± 4      |  17 ± 6        |  17 ± 6      |
> > | Puzzle-4×4         |   5 ± 2      |  16 ± 1      |  16 ± 2      |  12 ± 4        | **19 ± 5**   |
> > |--------------------|--------------|--------------|--------------|----------------|--------------|
> > | Average Score      |     44       |     45       |     39       |     65         | **67**       |
> >
> > Our contribution is orthogonal to diffusion or flow-matching policy methods: floq provides a flow-matching critic that can be combined with any policy learner. We also show this in **Appendix A.4 and Table 4**, where we found floq combined with ReBRAC gaussian policies substantially improves over monolithic critics (average score: **30 $\to$ 41**).
> >
> > If you have a specific comparison in mind, we would be happy to include it in the final version.
> >
> > ---
> >
> > ### > **It would be better to also add the flow-matching to preliminary to help readers understand the following method smoothly.**
> >
> > We have added a more thorough introduction to flow-matching to preliminaries (see **Section 3**). Thank you for this suggestion. Please let us know if you would like something else to be added.
> >
> > ---
> >
> > ### > **What is the TD-target in line 179.**
> >
> > The TD target we use is the quantity already defined in the paragraph:
> > $$
> > y(s, a) = r(s, a) + \gamma \frac{1}{m} \sum_{j=1}^m
> > \psi_{\tilde{\theta}}(1, \mathbf{z}'_{j} \mid s', a').
> > $$
> > This is the expected next-state Q-value under the target network, estimated by averaging $m$ flow-matched Q-value samples. The subsequent construction of the interpolant
> > $$
> > \mathbf{z}(t) = (1 - t)\mathbf{z} + ty(s,a)
> > $$
> > uses this $y(s,a)$ as the TD target to train the flow model. To avoid confusion, we will clarify this by explicitly referring to $y(s,a)$ as the ``TD target'' in the updated version.
> >
> > ---
> >
> > ### > **The authors state that Q_min and Q_max denote the minimal and maximal possible Q-value achievable on the task. But how do you calculate them in practice?**
> >
> > We obtain Q_min and Q_max by computing the minimum and maximum empirical returns in the offline dataset. During online learning, we calculate online returns and update Q_min and Q_max, similarly to previous works using categorical critics [1,2]
> >
> > [1] Bigger, Regularized, Categorical (BRC), High Capacity Value Functions Are Efficient Multi-Task Learners, Nauman et. al, 2025.
> > [2] Hyperspherical Normalization for Scalable Deep RL (SimbaV2), Lee et. al, 2025.
> >
> > ---
> > We are happy to answer any further questions. Thank you again for helping us to improve our manuscript. We hope that the clarifications paired with additional experiments increase your confidence in our work. If so, we kindly ask to update the score accordingly.

---

> > > ### Comment · Reviewer_eV3j · 2025-11-26
> > >
> > > I appreciate the author's efforts in the response and clarifications. This rebuttal addressed most of my concerns and thus I'm happy to increase my score.

---

### Official Review · Reviewer_H9C5 · 2025-11-01

**Soundness:** 3
**Presentation:** 3
**Contribution:** 3
**Rating:** 6
**Confidence:** 3

**Summary:**

The paper proposes Flow-matching Q-functions (floq). floq trains a time-dependent,state-action-conditioned velocity field $v_\theta(t,z|s,a)$ such that at the forward operator $\psi_\theta$ defined by the vector field equals $\psi_\theta(j/K, z | s, a) \ z + \frac{1}{K}\sum_{i=1}^j v_\theta(\frac{i}{K}, \psi_\theta(\frac{i-1}{K}, z| s, a) | s, a)$ with $Q(s, a, z) \triangleq \psi_\theta(1, z|s,a)$ and $z_0 \sim\text{Unif}[l,u]$. Furthermore, $z_t = (1-t) z_0 + t y(s,a)$ where $y(s,a)$ is the boopstrapped TD-target. The authors validate floq on 50 tasks from OGBench, demonstrating approximately 1.8× improvement over baseline FQL on challenging tasks, and show it scales better than ResNets or monolithic Q-function ensembles.

**Strengths:**

1. Strong ablation studies on floq
2. Comprehensive evaluations with superior performance in difficult tasks such as puzzle-4x4 and cude-double
3. Novel connection between flow matching and Q-learning
4. Nice analysis of design choices in section 4.3. It gives very inspirational analysis on how to design the floq architecture.

**Weaknesses:**

1. Lack of theoretical understanding
2. All implementations of floq build on FQL
3. The idea is similar to distributional RL, but it not widely compared
4. While design choices are grounded, they are not systematic but more results from trial-and-error

**Questions:**

1. For a fair comparison, I think one should do scalar critic + gaussian policy, scalar critic + flow policy, floq critic + gaussian policy and floq critic + flow policy. Is there any result on floq critic + gaussian policy?
2. floq's main contribution is on its critics. Is this applicable for other models such as REBRAC or only special to FQL?
3. In Figure 5, why do ResNets show performance degradation at high FLOPs (e.g., antmaze-giant drops from 45% at 16 FLOPs to 31% at 32 FLOPs)? More importantly, floq also shows this degradation pattern in Figure 4 (antmaze-giant: 86% at K=4 → 70% at K=8 → 55% at K=16). What is the fundamental cause of this degradation?
4. in Figure 6, floq achieves better performance than monolithic ensembles. Is ensemble also applicable for flow? If so, is there any benefit gain?
5. I am not 100% convinced why HL-Gauss encoding helps here. Could you explain more in details why this helps in this scenario?
6. The performance seems to depend heavily on hyperparameters. How easy/difficult is it to adopt the algorithm for a new test task?

---

> ### Author Response · Authors · 2025-11-23
>
> Thank you for your thorough review and a positive assessment of our work. We are happy that you appreciate the novelty of applying flow-matching to Q-learning and find our evaluations comprehensive. To address your concerns, we have added new experimental results comparing floq to distributional RL methods (IQN and C51), where we find that floq performs best. We have also added new results instantiating floq with Gaussian policies, on top of the ReBRAC method and again observe that floq performs better than using a monolithic critic. We also ran several experiments to understand the core reasons behind degradation in performance with a higher value of $K$ that we also discuss below. Finally, we provide clarifications to the other questions. We have updated the paper with these results as well.
>
> **Please let us know if our response addresses your concerns, and if so, we would be grateful if you are willing to raise your score.**
>
> ---
>
> ### > **The idea is similar to distributional RL, but it is not widely compared.**
>
> This is a good suggestion! Our experiments in **Table 2, Appendix A.3** in the submission already included comparisons to **IQN**, a performant and popular distributional RL method, and these results showed that **floq already outperforms these comparisons**. We now add a new comparison to **C51**, another representative distributional RL method and find that **floq also outperforms this approach substantially**. Our comparisons also reveal that IQN outperforms C51, indicating that the distributional RL comparison we had in the submission was a stronger baseline compared to other popular methods such as C51.
>
> | **Environment**        | **FQL**          | **FQL-IQN**      | **FQL-C51**      | **floq (Def.)**   | **floq (Best)**  |
> |--------------------|--------------|--------------|--------------|----------------|--------------|
> | Antmaze-Large      |  85 ± 4      |  86 ± 1      |  79 ± 3      | **94 ± 4**     | **94 ± 4**   |
> | Antmaze-Giant      |  14 ± 29     |   5 ± 6      |   0 ± 0      |  70 ± 8        | **86 ± 4**   |
> | Hmmaze-Medium      |  58 ± 25     |  27 ± 17     |  23 ± 16     | **98 ± 1**     | **98 ± 1**   |
> | Hmmaze-Large       |  14 ± 10     |  22 ± 5      |   6 ± 2      | **52 ± 8**     | **52 ± 8**   |
> | Antsoccer-Arena    | **49 ± 11**  |  44 ± 3      |  36 ± 7      |  49 ± 10       |  49 ± 10     |
> | Cube-Single        |  94 ± 5      | **98 ± 1**   | **98 ± 1**   |  **99 ± 2**        | **99 ± 2**   |
> | Cube-Double        |  29 ± 8      |  57 ± 2      |  40 ± 11     | **72 ± 15**    | **72 ± 15**  |
> | Scene-Play         |  78 ± 7      |  80 ± 4      | **82 ± 11**  |  **83 ± 10**       | **83 ± 10**  |
> | Puzzle-3×3         |  14 ± 4      | **20 ± 3**   |  13 ± 4      |  17 ± 6        |  17 ± 6      |
> | Puzzle-4×4         |   5 ± 2      |  16 ± 1      |  16 ± 2      |  12 ± 4        | **19 ± 5**   |
> |--------------------|--------------|--------------|--------------|----------------|--------------|
> | Average Score      |     44       |     45       |     39       |     65         | **67**       |
>
> **We have also added the C51 results to Table 2 in the Appendix A.3 in the paper.**
>
> We would also like to emphasize that although our flow Q-function outputs a distribution over Q-values (which we train to match to the Dirac-delta distribution), floq is **not** a distributional RL algorithm in the classical sense. Importantly, **we do not apply any distributional TD-backup operator** and instead use a simple expected value TD backup $y(s,a) = r + \gamma\mathbb{E}_{z'}[Q(z' \mid s', a')]$, obtained by averaging samples from the target floq critic. We agree that combining flow-based Q-functions with a distributional RL update (e.g., quantile regression or categorical projection) is an interesting direction, and plan to explore it in future work. We will add this discussion to the paper.

---

> ### Author Response · Authors · 2025-11-23
>
> ### > **Flow Critic + Gaussian Policy and Applicability To Other Methods Such As ReBRAC**
>
> Thank you for the question. To address this question, we ran new experiments evaluating the flow critic with a Gaussian policy. Concretely, we evaluated the efficacy of a **floq critic with ReBRAC** and show our results below. We observe that **floq (ReBRAC)** substantially improves over the vanilla ReBRAC approach, increasing the average score from **30% to 41%** indicating that floq is effective on top of other methods such as ReBRAC as well, that use Gaussian policies.
>
> | **Environment**        | **ReBRAC**        | **floq (ReBRAC)** | **FQL**           | **floq (FQL)**    |
> |-------------------|---------------|---------------|---------------|---------------|
> | Antmaze-Large     | 95 ± 4        | 97 ± 3        | 85 ± 4        | 94 ± 4        |
> | Antmaze-Giant     | 27 ± 22       | 91 ± 6        | 14 ± 20       | 70 ± 8        |
> | Hmmaze-Medium     | 16 ± 9        | 57 ± 12       | 58 ± 25       | 98 ± 1        |
> | Hmmaze-Large      | 2 ± 1         | 0 ± 1         | 14 ± 10       | 52 ± 8        |
> | Antsoccer-Arena   | 0 ± 0         | 1 ± 2         | 49 ± 11       | 49 ± 10       |
> | Cube-Single       | 92 ± 4        | 89 ± 4        | 94 ± 5        | 99 ± 2        |
> | Cube-Double       | 7 ± 3         | 3 ± 3         | 36 ± 6        | 72 ± 15       |
> | Scene-Play        | 50 ± 13       | 58 ± 8        | 78 ± 7        | 83 ± 10       |
> | Puzzle-3x3        | 2 ± 1         | 9 ± 6         | 16 ± 5        | 17 ± 6        |
> | Puzzle-4x4        | 10 ± 3        | 6 ± 4         | 11 ± 3        | 12 ± 4        |
> | **Average Score** | **30**        | **41**        | **45**        | **64**        |
>
>
> We have added these results to **Table 4 and Appendix A.4** in the paper.
>
> ---
>
> ### > **Implementation builds on FQL.**
>
> We acknowledge that our implementation builds on the FQL codebase; this choice was intentional to ensure a fair comparison. By building on FQL, we isolate the performance differences to our proposed flow-matching critic, ensuring that the observed performance improvements arise solely from improved Q-value learning rather than unrelated implementation factors.
>
> ---
>
> ### > **Degradation at high K.**
>
> This is a great question! Please note that we observe a degradation at a higher $K$ not just for floq but also for the monolithic ResNet approach (refer **Figure 6**). This makes us believe that a degradation in the critic performance is largely not specific to floq but is also applicable for a ResNet. We are still running several diagnostic experiments to evaluate floq, but we have some results that indicate that a degradation with larger $K$ stems from **overfitting to very high capacity targets** with both the ResNet and the floq architecture.
>
> For example, in Antmaze-Giant (refer **Figure 6**), the best-performing configuration **($K=4$) exhibits a higher TD error than the $K=8$ or $K=16$** settings, whose lower TD errors coincide with worse control performance. This aligns with the view that excessively small TD error often indicates overfitting rather than better value estimation. Supporting this, when we add a small amount of Gaussian noise to the TD target for the $K=16$ critic, performance improves by about $7$ to $10$ percent and the TD error increases to match the healthier regime of $K=4$. Taken together, these results point to a shared underlying cause: very high $K$ increases the critic capacity in a way that encourages overfitting to sharp targets, and moderate slack or noise helps maintain the robustness needed for strong downstream performance.
>
> ---
>
> ### >  **Lack of Theoretical Understanding.**
>
> Thank you for the suggestion. You are right that we do not provide a new theoretical result, but this is because our method does not modify the theoretical foundations of TD learning in the first place. floq still utilizes an expected Q-value backup, just like standard deep Q-learning methods such as DQN and SAC. When deep neural networks are used as function approximators, we are unaware of any theoretical convergence guarantees for TD learning in these methods, and the same limitations apply to floq.
> The key contribution of floq is to replace the standard monolithic neural network parameterization of $Q(s,a)$ with a flow-matching formulation that **enables iterative computation of the Q-value**. The underlying TD objective, Bellman operator, and convergence properties remain unchanged, so existing theoretical results (and the lack of such results) for deep RL continue to apply. In this sense, floq is a representational enhancement of Q-learning rather than a departure from its established theoretical framework.
> That said, if there is a specific theoretical result you would like us to demonstrate, we would be happy to attempt it within the remaining rebuttal period.
>
> ---

---

> ### Author Response · Authors · 2025-11-23
>
> ### > **Ensemble of Flow Matching Critics**
>
> floq is parallelizable in the same way as standard monolithic critics. In fact, in the paper, we already trained an **ensemble of 2 floq critics**. Increasing the ensemble size of floq critics should yield performance improvements analogous to increasing ensembles of monolithic critics.
>
> Importantly, our ablation comparing floq to ensembles of monolithic critics demonstrates that even when giving monolithic critics increased parallel compute (e.g., 4× or 8× ensembles), floq still substantially outperforms them. Thus, the gains from parallel compute of monolithic critics do not close the performance gap observed in our experiments.
>
> ---
> ### > **HL-Gauss Encoding**
>
> We thank the reviewer for the question. As discussed in **Appendix A.6 (point 3)** and shown in **Figure 14**, HL-Gauss embeddings provide a significant advantage over using the raw or normalized scalar values. The interpolant inputs $z(t)$ are inherently non-stationary because they depend on Q-values that evolve over the course of training. **HL-Gauss embeddings with broader bin coverage reduce the network’s sensitivity to these non-stationary inputs**, stabilizing training and improving performance. **Figure 14 (right)** shows that increasing the HL-Gauss sigma, which increases the bin coverage, consistently improves performance.
>
> We also saw that using the HL-Gauss encoding often reduces the gradient norm and stabilizes training. For eg. in HM-Large (Task 5), it reduced roughly $4\times$ (from $10^5$ to $2.5 \times 10^4$) (see **Appendix A.6, point 3** and **Figure 15**).
>
> ---
>
> ### > **Hyperparameter Dependency**
>
> While we do introduce a small number of additional hyperparameters beyond FQL, we follow **exactly the same tuning protocol prescribed by OG-Bench and used in prior work**: select the hyperparameter value on the default task for each environment and keep it fixed for all remaining tasks. We also observe that **even when the floq hyperparameters are kept frozen across all tasks**, floq (default) in **Table 1** still outperforms FQL. This indicates that practitioners should not find floq more difficult to tune in practice.
>
> We also report per-environment tuning of floq hyperparameters (**floq (best) in Table 1**); where **per-environment tuning yields only marginal overall gains** ($56 \to 59$, see Table 1), which we interpret as evidence of robustness (if per-task tuning produced large wins it would indicate brittleness, but it does not). Practically this means users can adopt floq on new tasks by using the same FQL defaults plus our floq defaults (we have reported the default hyperparameters in **Table 6** in the Appendix), and if further gains are desired, prioritize (1) the number of integration steps, then (2) the initial noise range. We’re happy to add an expanded discussion to the supplementary material to make these guidelines explicit. Our ablations in **Appendix A.6** are already meant to describe how different design choices affect performance and what are thumb rules to tune them.
>
> ---
>
> We are happy to answer any further questions. Thank you again for helping us to improve our manuscript. We hope that the clarifications paired with additional experiments increase your confidence in our work. If so, we kindly ask to update the score accordingly.

---

> > ### Author Response · Authors · 2025-11-28
> >
> > Dear Reviewer H9C5,
> >
> > Thank you again for taking the time to review our paper. We wanted to kindly follow up to ask whether our rebuttal addressed your questions and concerns, or if there is anything further you would like us to clarify. We are happy to provide any additional details that may help with your assessment.
> >
> > If our responses resolved your concerns, we would greatly appreciate it if you could update your score accordingly.
> >
> > Thank you again for your time and consideration.

---

### Official Review · Reviewer_GXf4 · 2025-11-01

**Soundness:** 3
**Presentation:** 4
**Contribution:** 3
**Rating:** 8
**Confidence:** 4

**Summary:**

This work aims to improve scalability of Offline RL by introducing a flow model based Q-critic. The most closely related past work is FQL (ICML2025), which included a flow-based policy, but a regular Q-critic; the current work uses the same policy type, but adds a flow-model critic. The model works by uniformly sampling an initial noise value in the interval [Q_min, Q_max], then applying Euler integration steps to the noise with a learned velocity field for a few steps to end up at the final Q estimate. (The velocity field is also conditioned on the state and action.) Their architecture is called *floq*. Another major point in their discussion is the need for intermediate targets when training the flow which is done by a linear interpolant. The representation of the interpolant inputs is another major design choice: they use a categorical representation based on the HL-Gauss method from Farebrother (2024), and they also add a Fourier basis representation for the time t. Experimentally, they evaluate their method on OGBench (the same major testbed as used by FQL (ICML2025)), and consistently outperform FQL in terms of performance. Several ablation and sensitivity studies show the necessity of the different design choices in the current configuration, and how they alter the performance.

**Strengths:**

- Well written, and well presented.

- Thorough experimentation. The work provides several sensitivity studies and much material for readers to examine regarding the performance. This can be a good reference for reproducing the results or building on the results.

- Performance is compared on the same settings as FQL, which is an established method, the FQL results in the current paper match the published results, and the method of the current paper outperforms it clearly.

- Some of the discussion was interesting, e.g., about the  usefulness of iterative computation for scaling in deeplearning, and how intermediate targets in such cases tend to be helpful.

**Weaknesses:**

- The computation time increases. The runtime in Appendix A.8. shows that it is quite a bit slower than FQL (up to around 7 times slower for 16 integration steps). This may reduce the practicality of the method.

- Many design choices appear ad hoc to me (e.g., using HL-Gauss, and other design choices). These appear mostly engineered by testing things, without a strong theoretical backing for the choices. Such advances are also useful, but it is not clear to me whether these are likely to stand the test of time, so I list it as a weakness.

**Questions:**

I have no major question, but feel free to respond to my perceived weaknesses.

A few typos:

Line 138: “the our” change to “our”

The definitions of the v and q functions around lines 126 and 127 appear wrong. You add the factor $1/(1-\gamma)$ in front, but at the same time, you added the $\gamma^t$ inside the sum of the expectation. The original definition of these is: $\mathbb{E}[\sum_{t=0}^\infty \gamma^t r_t]$, so there will not be a $1/(1-\gamma)$ factor in front. You get this factor, if you instead define the discounted state visitation distribution, and define the objective as the expected reward under the normalized discounted state visitation distribution (where the discounting appears in the distribution, not as a multiplier to the reward).

---

> ### Author Response · Authors · 2025-11-23
>
> Thank you for the review, your positive feedback, and helpful suggestions. We are very happy that you find our experiments thorough and like the presentation of our results. We would also like to point your attention to additional experiments that we have added to the paper. In these experiments, we validate that our proposed method can be successfully used when paired with different base algorithms beyond FQL, such as ReBRAC. We also expand our analysis of training behavior under different numbers of integration steps. These experiments are detailed in the updated manuscript.
>
> **Please let us know if your questions are addressed.**
>
> ---
>
> ### > **Design choices are adhoc**
>
> You are right that we did test several choices extensively to understand what works well in practice. However, we believe that a number of these choices are well motivated by prior evidence.
>
> For example, as discussed in our response to Reviewer H9C5, we found that using a scalar encoding of the interpolant input led to a large gradient norm due to non-stationarity, whereas a categorical encoding provided a magnitude-invariant representation. We instantiated this using HL-Gauss, motivated by prior work (Farebrother et al., 2024), which has shown benefits in scaling model capacity. We agree that other magnitude-invariant encodings could also work, but we believe the **core principle—using a magnitude-invariant encoding—is justified and useful**.
>
> Likewise, regarding the distribution of the initial noise sample, we think that ensuring curved flows and sufficient capacity usage is important—an aspect not present in prior work on images. We chose to do this using a higher-variance initial distribution. While other mechanisms could be explored in the future, the **core principle—using approaches that improve flow curvature—remains central**.
>
> We agree that additional theoretical grounding for an approach instantiated on these principles would be valuable, but we are unaware of prior theoretically-motivated solutions to these challenges. Finally, we emphasize that our primary goal was to demonstrate that a flow-matching critic can outperform standard monolithic architectures, and our design choices reflect a simple and practical mechanism to achieve this.
>
> ---
>
> ### > **Computation time**
>
> Indeed, increasing the number of integration steps introduces additional computation. However, an important finding is that floq does not require large step counts to outperform FQL. With values such as **K = 4**, the runtime overhead is modest while still yielding clear performance improvements. In this regime, the added cost is effectively amortized by the gains in performance. We will clarify this point explicitly in the main text and add guidance recommending small K as a default for new tasks.
>
> We also note that this runtime pattern is **not unique to the critic**. Moving from a Gaussian policy to a flow-based policy (as in recent works) incurs a similar multiplicative cost from integrating the flow ODE. Thus, the computation needed for a flow critic is comparable to overheads already present when replacing a standard policy parameterization with a flow-based one. We will add this discussion of computation time to the paper.
>
> ---
>
> ### > **V and Q functions in line 126 and 127**
>
> Thanks for pointing this out — we have removed the extraneous $\frac{1}{1-\gamma}$ factor.
>
> ---
>
> We hope that the above clarifications, together with the expanded experiments described in the general response, increase your confidence in our work. Thank you again for your help in improving the manuscript.

---

> > ### Comment · Reviewer_GXf4 · 2025-11-27
> > **My questions are addressed**
> >
> > I appreciate the response.
> >
> > I see your point of view that lower K can already improve performance. At the same time, sometimes you do need larger K, such as K=8 (e.g., in Figure 4, Cube Double). I think there are still some tradeoffs regarding compute time and performance. This is natural, so I don't view it as a major issue either way.
> >
> > A small comment about the updated manuscript is that there are some equation references with ?? as the pointer. Also, perhaps you didn't notice, but the update during the rebuttal allows for an extra page (up to 10 pages). This doesn't affect my assessment.
> >
> > Overall, I think my score is already good, and the paper is strong, so I would like to keep my score.

---

> > > ### Author Response · Authors · 2025-11-28
> > >
> > > Thank you for pointing out the issue with the unresolved equation references in the updated manuscript. We have now carefully reviewed the paper and fixed all equation references that previously appeared as “??”.
> > >
> > > Thank you again for your feedback, positive assessment and for keeping your strong score, we sincerely appreciate it.

---

### Meta-Review · Area_Chair_ymod · 2026-01-08

**Summary:**

This work improves offline RL by introducing floq. Floq extends FQL (Park et al., 2025) by incorporating a Q-network based on flow matching. By learning a velocity field, floq enables iterative refinement of Q-estimates. This approach not only provides denser supervision during training but also allows for test-time compute scaling. Evaluated on the challenging OGBench suite, floq consistently outperforms FQL and other baselines. With reviewers satisfied by the rebuttal and the paper's strong empirical validation, this is a clear accept.

**Reviewer Concerns:**

Initially the reviewers raised several concerns as below:
- the computational time increases
- lacking comparison with distributional RL
- design choices are ad-hoc
- lacking theoretical understanding
- the experiments are only with FQL

In the rebuttal, the authors successfully defend or address them through additional experiments.

**Reviewer Scores:**

The initial scores are 8664.  The score-4 reviewer has increased their score to 6.  I believe one or both score-6 reviewers might increase their score to 8.

---

### Decision · Program_Chairs · 2026-01-26

Accept (Poster)